# NorQD AAA+ complex drives metal insertion by a twisting mechanism

Maximilian Kahle [1,2,5], Sofia Appelgren [2,3,5], Finja König[2], Marta Carroni [2,4], Pia Ädelroth [2] ✉ & Petra Wendler [1] ✉

ATPases associated with diverse cellular activities (AAA+ -ATPases) catalyse a wide range of remodelling events in all phyla. AAA+ -ATPases of the MoxR-like family typically co-operate with von Willebrand factor type A (VWA) domain containing proteins to facilitate target remodelling and metal ion insertion, but their mechanism of action is poorly understood. We studied the bacterial AAA+ -ATPase NorQ in complex with its VWA domain partner protein NorD, which are essential for nitric oxide reductase (NOR) activity. Our cryo-EM structures and biochemical analyses show that NorQ and NorD engage through two key interfaces: (i) a finger-like extension protruding from the VWA domain that penetrates the central pore of the NorQ hexamer, and (ii) the NorD C- terminus, which contacts the post-sensor 1 loop of NorQ. Our data reveal that NorQ activity remodels a linker region in NorD essential for metal insertion. Together, these findings support a model in which the NorQ complex exerts a twisting and stretching force on the NorD linker, thereby enabling metal insertion into its target NOR.

Metalloproteins constitute about 30% of all known proteins, and half of all enzymes contain metal cofactors[1]. The insertion of the correct metal ion into the active site of metal containing enzymes is crucial for their functionality and often depends on metallochaperones. Nevertheless, the mechanism of metal insertion and the interaction between metalloprotein and chaperone are not yet fully understood. Denitrification, i.e. respiration of nitrate, is performed by soil-living bacteria and is dependent on a set of highly metalated enzymes, reducing nitrate into dinitrogen in four separate enzymatic steps. The third reduction is performed by the haem-copper oxidase nitric oxide reductase (NOR), reducing nitric oxide into nitrous oxide. Cytochrome $c$-dependent NOR ($c$NOR) is a membrane protein, consisting of two subunits: NorB (52 kDa), and NorC (17 kDa), holding several cofactors: three haems and one non-haem iron ($Fe_B$)[2-4]. The $c$NOR gene cluster contains six genes: structural genes $norC$ and $norB$, chaperone encoding genes $norQ$ and $norD$, and genes $norE$ and $norF$, encoding small membrane proteins of unknown function. We have previously

shown that the insertion of $Fe_B$ in $c$NOR from *Paracoccus (P.) denitrificans* is dependent on the two chaperone proteins NorQ and NorD, and that $c$NOR is inactive without $Fe_B$[5].

NorQ (30 kDa) belongs to the large family of AAA+ ATPases[6,7], and shares several conserved features with AAA+ protein family members. AAA+ proteins typically form ring-shaped hexamers that thread the substrate protein in an ATP dependent manner through their central pore and thereby remodel it. This is often done in a hand-over-hand mechanism[8,9], whereby the protomers are arranged in a right-handed spiral staircase. The ATP bound, topmost protomers are bound to the substrate, while the subunit at the spiral seam usually has no nucleotide or substrate bound. Upon ATP binding, the seam subunit will relocate to the top, attaching to the substrate and pulling the amino acid chain further into the pore while remodelling it[9]. NorQ belongs to the Pre-Sensor 2 Insert Clade (clade 7) of AAA+ proteins, which is formed by MoxR-like proteins such as dynein, CbbQ, and RavA[10]. Their defining feature is a β-hairpin insert on helix 2 (H2i) near the substrate

[1]Institute of Biochemistry and Biology, Department of Biochemistry, University of Potsdam, Potsdam-Golm, Germany. [2]Department of Biochemistry and Biophysics, Stockholm University, Stockholm, Sweden. [3]Department of Biology, Philipps University Marburg, Marburg, Germany. [4]Swedish Cryo-EM Facility, Science for Life Laboratory Stockholm University, Solna, Sweden. [5]These authors contributed equally: Maximilian Kahle, Sofia Appelgren. ✉e-mail: pia.adelroth@dbb.su.se; petra.wendler@uni-potsdam.de

binding loop. MoxR-like AAA+ proteins often involve a VWA domain partner protein to perform their remodelling activity[6], but very little is known about the detailed mechanism of their interaction. The 69 kDa protein NorD is the VWA domain partner of NorQ. Its VWA domain contains a conserved metal ion dependent adhesion site (MIDAS), consisting of five residues (DxSxS-T-D) coordinating a divalent metal ion, presumed to be $Mg^{2+}$. As the name implies, the metal ion provides a binding site for other proteins, with the target protein contributing a sixth ligand, adhering to the metal ion and thus to the VWA domain protein[11]. The MIDAS motif is suggested to provide the binding between the chaperone complex and its final substrate[12]. Even though MoxR-like AAA+ ATPases are strictly prokaryotic, interaction between AAA+ ATPases and VWA domain proteins have previously been observed also in eukaryotes, for example, in Mg-chelatases, inserting magnesium into chlorophyll[13] and Rea1, involved in the remodelling of pre60S ribosomes in yeast [14].

Here, we present cryo-EM structures of different NorQD complexes from *Paracoccus denitrificans* and *Jhaorihella thermophila* which in combination with mutagenesis studies, enzyme activity data and AlphaFold predictions, shed light on the function of the NorQD complex. We show that a finger motif protruding from the VWA domain binds in the pore of NorQ, and that this interaction is needed for *c*NOR activation. Finally, we propose a tentative model for how structural changes in *c*NOR are induced by the ATPase function of the NorQD complex.

## Results

### Domain architecture and structure of *Pd*NorQ

We collected a cryo-EM dataset of the NorQ^WB AAA+ complex from *Paracoccus denitrificans* (*Pd*) without NorD or ATP present. The NorQ subunits of this complex carry a mutation (E109Q) in the Walker B motif (WB), leading to low ATPase activity but retained ATP binding, which stabilises complex formation[12]. Since the 2D classes showed highly preferred orientation (Supplementary Fig. 1a), we combined these data with a dataset collected at a tilt angle of 30 degrees to yield a map suitable for modelling (Fig. 1a, Supplementary Fig. 1b). The map was resolved to 3.7 Å overall and up to 3.3 Å in the core of the AAA+ module enabling us to determine the nucleotide in the binding pockets unambiguously (Supplementary Fig. 1c–e, Supplementary Table 1). The *Pd*NorQ^WB hexamer forms a closed, right-handed spiral, in which the topmost three subunits A, B and C are bound to ATP and the two bottom subunits D and E as well as the seam subunit F are bound to ADP (Fig. 1a, Supplementary Fig. 2a). All helix 2 inserts (H2i) are long β-hairpins which arrange to form a 15 Å pore on the N-terminal side of the hexamer (Fig. 1a). Each pre-sensor 1 (preS1) loop forms a β-hairpin that is positioned at the subunit interface to the right-hand neighbour in the ring. Two more loops, the classical pore loop and the post-sensor 1 (postS1) loop, are lining the inside of the pore through the AAA+ complex, with the postS1 loop being closer to the C-terminal end of the pore (Fig. 1a).

### Binding mode of *Pd*NorQ^WB to *Pd*NorD^VWA

We also wanted to investigate the structure of the entire *Pd*NorQD complex, but cryo-EM experiments with this complex resulted to an insurmountable preferential orientation of the particles in the ice, suggesting an even stronger hydrophobic interaction of *Pd*NorD than *Pd*NorQ with the air-water interface. However, binding only the *Pd*NorD VWA fragment (*Pd*NorD^VWA) to *Pd*NorQ overcame this limitation and we solved the *Pd*NorQ^WBD^VWA structure without ATP in the buffer by cryo-EM (Fig. 1b, Supplementary Fig. 3, Supplementary Table 1). The nucleotide occupancy of this complex is similar to that of the *Pd*NorQ^WB complex (Supplementary Fig. 4a–c), except that subunit F at the seam is lacking the ADP. We determined the *Pd*NorQ^WBD^VWA structure under oxidizing conditions and with dithiothreitol (DTT) under reducing conditions, achieving resolutions of 3.2 Å and 3.8 Å,

respectively (Supplementary Fig. 3 and 5). The overall root mean square deviation (RMSD) between the Cα atoms of the two models derived from maps obtained under these respective conditions is 1.075 Å across all 1525 amino acids of the AAA+ ring (Supplementary Fig. 4e). Under oxidizing conditions, we identified a disulphide bridge between cysteines 46 and 68 in the subunit F (Supplementary Fig. 4d), which distorts the nucleotide binding P-loop of the ATPase and prevents ATP from binding to this subunit. Apart from the disulphide bridge, the two maps are identical at the resolution obtained. The ATPase activity of the complexes containing the disulfide bridge is very low (Supplementary Table 2), which also explains the low activities observed previously[12]. We conclude that the formation of the disulphide bridge under the oxidizing conditions of purification without DTT, practically abolishes ATP hydrolysis and traps the complex in place.

In the *Pd*NorQ^WBD^VWA structure *Pd*NorD^VWA forms a globular domain located in the C-terminal, concave opening of the AAA+ ring. As previously predicted[12], a finger-like domain is stretching into the pore to contact the H2i at the N-terminal end of the pore. Its fingertip intercalates in the seam of the ring between subunits A and F, shifting the large AAA+ subdomain of subunit F by 10 Å towards subunit E and opening up the nucleotide binding site (Supplementary Fig. 6a). While the local resolution in the AAA+ domains in the oxidized *Pd*NorQ^WBD^VWA consensus map is around 3 Å, it drops to an average of 4 Å in the VWA domain (Supplementary Fig. 3c). Extensive 3D sorting of the data shows that NorD^VWA moves slightly around two attachment points in the pore of the AAA+ ring (Supplementary Movie 1): the H2i loops and the postS1 loop of subunit E. Binding of *Pd*NorD^VWA to *Pd*NorQ^WB also induces the slightly steeper spiral staircase in the AAA+ ring, leading to an increased total rise of ~7 Å between subunits A to D (Supplementary Fig. 6b, 6c). We suggest that the intercalation of the finger-like domain at the spiral seam with simultaneous binding of the VWA domain to (at least) subunit E primes the complex for remodelling action.

To verify whether the *Pd*NorD^VWA fragment retains similar *Pd*NorQD activity to full-length *Pd*NorD, we measured the ATPase activities of different *Pd*NorQD complex variants. The *Pd*NorQ complex has an ATPase activity of $13 \pm 1 \, min^{-1}$ hexamer$^{-1}$, which in complex with full length *Pd*NorD under reducing conditions increases to $18 \pm 3 \, min^{-1}$ complex$^{-1}$, and with *Pd*NorD^VWA is slightly lower at $10 \pm 0.7 \, min^{-1}$ complex$^{-1}$. Note that these values differ from previously published activity data[12], see above. In the complexes with the *Pd*NorQ Walker B mutation, these ATPase activities are severely inhibited but measurable (Supplementary Table 5). ATP exchange and binding can therefore take place despite the Walker B mutation. Structurally, we also observe this when we incubate *Pd*NorQ^WBD^VWA with ATP (Supplementary Fig. 7), as we obtain mainly hexameric and pentameric *Pd*NorQ arrangements that carry ATP in each visible nucleotide binding pocket (Supplementary Fig. 2c, 2d). The fully ATP loaded, hexameric and pentameric complexes display a significantly steeper spiral staircase with an increased total rise of ~16 Å between subunits A to E and stronger seam than seen in the *Pd*NorQ^WB complex without ATP addition (Supplementary Fig. 2a, 2b, 2c, 6c). We conclude that binding of ATP to subunits E and F of the *Pd*NorQ^WBD^VWA complex makes the connection between subunits A and F of the seam so flexible that subunits can be lost. The steep spiral arrangement of subunits D to F impairs *Pd*NorD^VWA binding, although the *Pd*NorQ^WBD^VWA plus ATP dataset also contained a subset of hexamers bound to the VWA domain of NorD. In these cryo-EM structures, the central pore is filled with density that must derive from *Pd*NorD^VWA, but the local resolution of this density is insufficient to make out secondary structure elements.

### The *Pd*NorD^VWA–*Pd*NorQ^WB interactions and their effects on ATPase activity

The local resolution at the contact points between *Pd*NorD^VWA and *Pd*NorQ^WB is ~3.5 Å in the oxidized *Pd*NorQ^WBD^VWA consensus map and

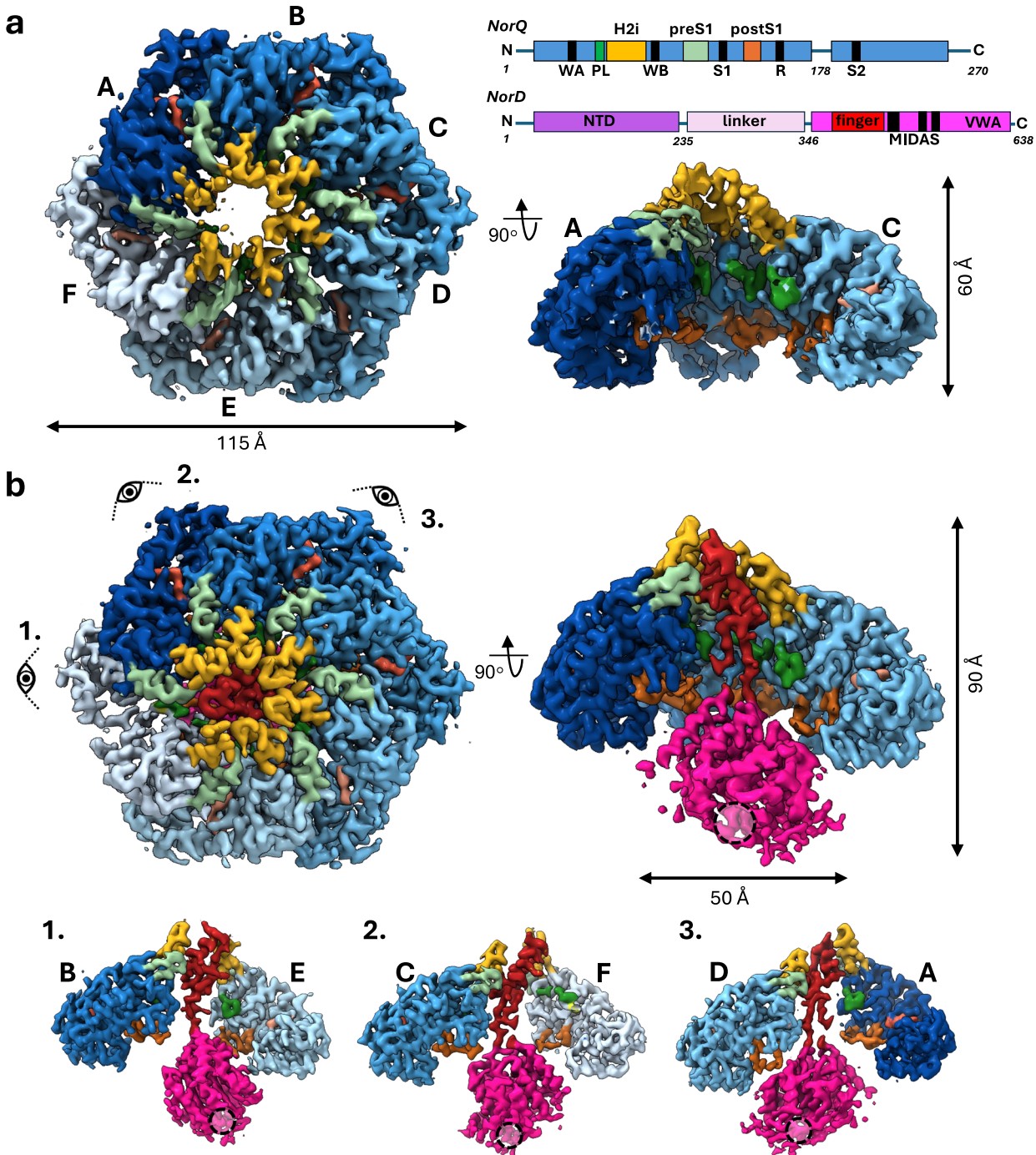

**Fig. 1 | Structural analysis of *Pd*NorQ^WB and *Pd*NorQ^WB^D^VWA. a** Cryo-EM structure of *Pd*NorQ^WB at 3.7 Å resolution shown from the N-terminal end and as a cut open side view, missing subunits D-F. The colour scheme corresponds to the domain architecture diagram on the right, illustrating the positions of the following motifs: WA (Walker A), PL (pore loop, dark green), H2i (helix 2 insert, yellow), WB (Walker B), preS1 (pre-sensor 1 loop, light green), S1 (sensor 1), postS1 (post-sensor 1 loop, orange), R (arginine finger), S2 (sensor 2), NTD (N-terminal domain, purple), VWA (von Willebrand factor type A, pink), VWA finger (red) and MIDAS (metal ion dependent adhesion site). The individual subunits of NorQ are colour-coded in shades of blue according to their position in the spiral, from top (dark) to bottom (light). **b** Cryo-EM structure of *Pd*NorQ^WB^D^VWA at 3.2 Å resolution viewed from the same directions as in a. The bottom row presents central slices of the complex from the three specified viewing angles, displaying only NorD and two NorQ subunits. The dashed circle indicates the location of the MIDAS site on the NorD^VWA domain.

can be interpreted in detail. The alpha helix of the finger motif of *Pd*NorD^VWA forms three hydrogen bonds to the H2i of subunit A of the *Pd*NorQ^WB hexamer, while the adjacent downstream loop segment forms two hydrogen bonds to subunit F (Fig. 2a, b, Supplementary Table 3). The *Pd*NorQ^WB^D^VWA interaction is further reinforced by several direct interactions involving conserved residues in the H2i of 5 out of the 6 AAA+ domains (Figs. 2a and 2c). Most prominent residues

involved in the interactions are Arg80, Tyr81, Leu83 and Thr88 in the conserved GRXLL/IXXXXT motif of the H2i (Fig. 2a, c, Supplementary Table 4). The interaction between the C-terminus of *Pd*NorD^VWA and the postS1 loop of subunit E is mediated by three hydrogen bonds, one of which is between backbone oxygen and nitrogen atoms (Fig. 2d, Supplementary Table 3). We furthermore observe a strong hydrophobic interaction between residues A593/L556/L636/V637/A638 of

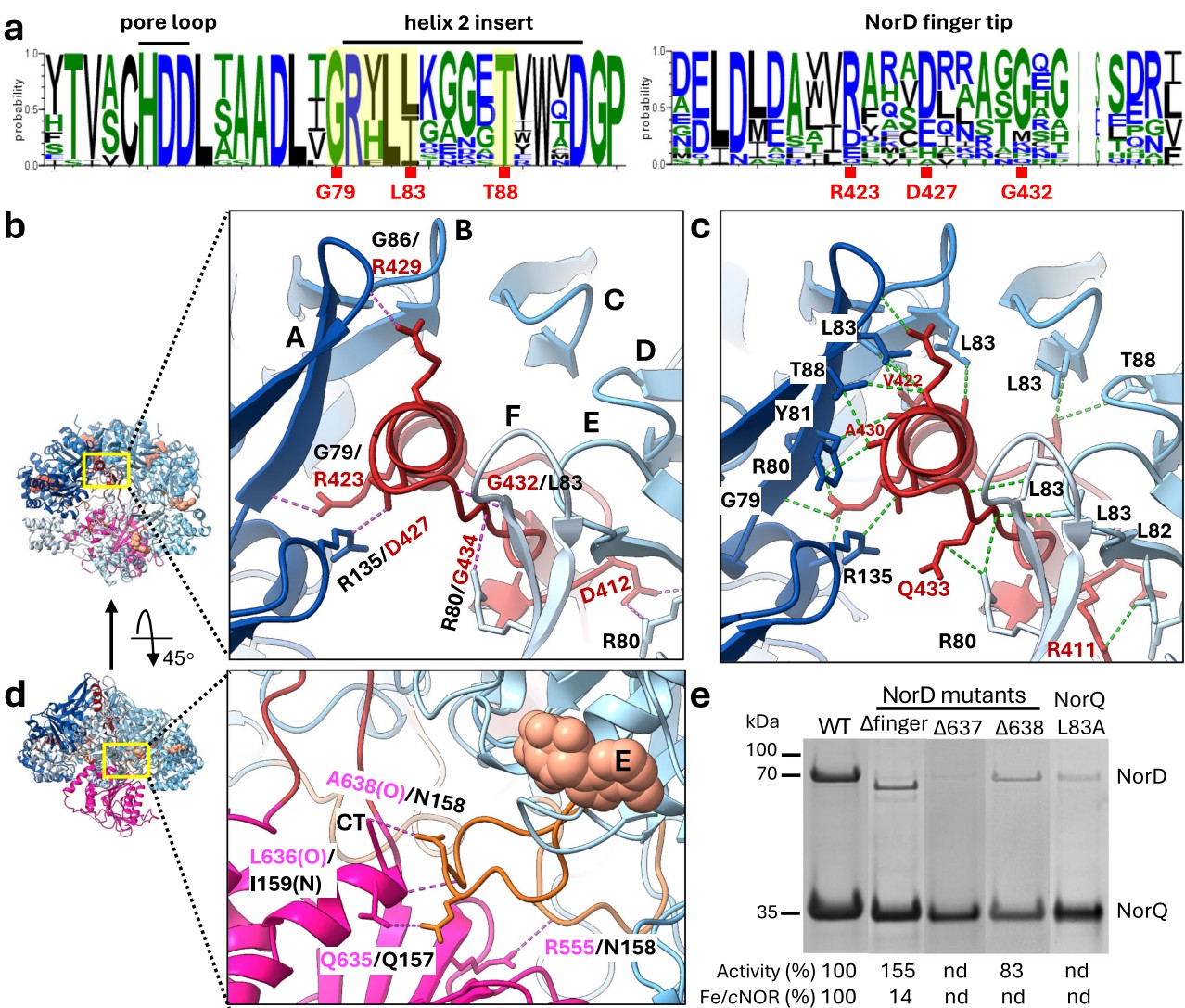

**Fig. 2 | Interactions between *Pd*NorQ^WB and *Pd*NorD^VWA. a** Sequence logos of pore loop, H2i (helix 2 insert) and NorD fingertip. Exemplary residues involved in NorQD interactions are highlighted in red. The GRXLL/IXXXXT motif is highlighted in yellow. **b** Hydrogen bonds between the NorQ H2i and the NorD VWA fingertip indicated by pink dashes. The participating residues are shown and listed. Individual chains of NorQ are labelled with capital letters A through F. **c** Contacts between the NorQ H2i and the NorD VWA fingertip with van der Waals overlap ≥ −0,4 Å are indicated as dashed lines and interacting residues are labelled. Details of contacts are described in Supplementary Tables 3 and 4. **d** Hydrogen bonds

between the NorD C-terminus (CT) and the NorQ postS1 loops. **e** Copurification of *Pd*NorD with *Pd*NorQ from wild type and from cells that carried deletions of the finger motif (Δfinger) or at the C-terminus (Δ637, Δ638) of NorD or a L83A mutation in NorQ. The percentage of ATPase activity compared to the wild type complex is provided for the Δfinger and Δ638 complexes (n = 3), and the relative amounts of iron insertion into *c*NOR (compared to wild type NorD) is given for the Δfinger construct (n = 2). nd = not determined. Statistical details are found in Supplementary Tables 5 and 6. Source data are provided as a Source Data file.

*Pd*NorD^VWA and I159/L160 of *Pd*NorQ, which will be discussed in more detail below. Interestingly, the postS1 loops of the ADP loaded subunits of the *Pd*NorD^VWA -associated AAA+ hexamer are more structured than those of the AAA+ ring alone, suggesting stabilization of the otherwise flexible loops by the interaction with the VWA domain (Supplementary Fig. 4a–c).

The importance of the interaction between the C-terminus of *Pd*NorD and the postS1 loop for complex stability was investigated by deletion of the very last (638), as well as the last two (637 and 638) residues of *Pd*NorD. The deletion of 638 led to a lower efficacy of co-purification, and the deletion of both, 637 and 638, abolished co-purification completely (Fig. 2e, Supplementary Fig. 8). Mutation of the L83 in the *Pd*NorQ H2i also lowered the efficacy of co-purification, as did deletion of the NorD^VWA finger (residues 403 to 448 exchanged for GGSIDGGS), however, not as much as the NorD 637 and 638 deletions. This suggests that the contact mediated by

the C-terminus of *Pd*NorD is stronger than that of the finger with the H2i, indicating that the integrity of the *Pd*NorQD complex is more dependent on this interaction. Moreover, the finger is not essential in stimulating ATPase activity, as ATP hydrolysis was increased to 28 ± 3 min⁻¹ complex⁻¹ in the complex without the finger (Fig. 2e, Supplementary Table 5). However, in the context of target remodelling, the *Pd*NorD finger is essential, as no non-haem iron was inserted into the *c*NOR protein when expressed with NorD lacking the finger (Fig. 2e, Supplementary Table 6). We conclude that binding of the VWA domain via its C-terminus to *Pd*NorQ might stimulate nucleotide exchange in the hexamer, while the finger motif is needed for *Pd*NorQ to perform remodelling action on *c*NOR, the substrate of *Pd*NorQD. Point-mutations in the *Pd*NorQ postS1 loop did not affect co-purification but resulted in a somewhat lower ATPase activity (Supplementary Table 5, Supplementary Fig. 8c).

### Domain architecture of $Jt$NorQ$^{WB}$NorD and the role of the NorD linker

All cryo-EM experiments involving the N-terminal and linker region of $Pd$NorD in the complex lead to a strong preferential orientation of the particles in the ice, suggesting a possible hydrophobic interaction of these regions with the air-water interface[12]. Indeed, we find stretches of hydrophobic amino acids in the beginning of the linker region of $Pd$NorD, which are less pronounced in *Jhaorihella thermophila* (*Jt*) NorD, a variant of the protein from a thermophilic bacterium. Furthermore, Blue native PAGE analysis confirms a higher stability of the $Jt$NorQD complex in comparison with the $Pd$NorQD complex (Supplementary Fig. 8b and 8d). ATPase activities of the $Jt$NorQ and $Jt$NorQD are similar to the $Pd$ complexes (Fig. 3a, Supplementary Table 5). Due to all these considerations, to obtain a full structure of the NorQD complex including also the NorD N-terminal part, we imaged and obtained structures of the $Jt$NorQ$^{WB}$D complex (Fig. 3b). These experiments allowed us to identify 4 conformational states of the $Jt$NorQ$^{WB}$D complex, which were resolved with a mean resolution of 3.2 Å to 3.9 Å (Fig. 3c, Supplementary Figs. 9–11). These maps also show signs of anisotropy due to preferential orientations, but they allow localization of secondary structure elements. By far the most populated subset of the dataset (state 1) shows densities for the NorQ$^{WB}$ hexamer and the VWA domain of NorD. A Phenix-refined model of this map can be superimposed on our model of $Pd$NorQ$^{WB}$D$^{VWA}$ with an RMSD of 0.687 Å across 1483 pruned $C_\alpha$ atom pairs (Supplementary Fig. 11c), indicating a very high structural similarity of this complex between the two species (sequence identity: 81% for NorQ and 65% for NorD). Even though less well resolved overall than the $Pd$NorQ$^{WB}$D$^{VWA}$ complex, the $Jt$NorQ$^{WB}$D dataset showed three states with a density for the N-terminal domain of NorD, which is bound either to the AAA+ subunit to the left of the topmost ATP-bound subunit (state 2), called subunit B in our study, or to the lowest ATP-bound subunit in the spiral arrangement (state 3 and 4), subunit C (Fig. 3c, Supplementary Fig. 11). The binding between the N-terminal domain of NorD and the small alpha helical subdomain of the ATPase subunit appears to be mediated by hydrophobic interactions. Interestingly, the hydrophobic region of the N-terminal domain of NorD that interacts with the C-terminal helices of NorQ, is covered by parts of the linker that connects the N-terminal domain with the VWA domain in the AlphaFold model of NorD (Supplementary Fig. 12a–c). These linker residues 242-393 of NorD are not resolved in any of our $Jt$NorQ$^{WB}$D maps (Supplementary Fig. 11). The arrangement of the VWA domain finger motif is identical in all 4 states of $Jt$NorQ$^{WB}$D, but the orientation of the globular VWA domain harbouring the C-terminus of NorD differs between states 1–3 and state 4 (Fig. 3c, Supplementary Fig. 12d). While the unresolved linker region in state 3 spans a distance of 67 Å between A393$^{VWA}$ and V241$^{NT}$, it is 53 Å in state 4, and 45 Å in state 2 (Fig. 3d). Presumably these changes in distance spanned by the linker region are involved in the $c$NOR remodelling mechanism of the NorQD complex (see "Discussion"). To investigate the functional significance of the NorD linker, we expressed $Pd$-$c$NOR from constructs with different parts of the NorD linker deleted (Fig. 3e and Supplementary Fig. 13). These constructs all led to non-functional $c$NOR (Fig. 3e) lacking the Fe$_B$, supporting that both the length and specific regions of the linker are important for target remodelling. Expression and complex formation of the $Pd$NorQD was verified in a construct with the entire linker deleted (Δlinker or NorDΔ$^{236-342}$) (Supplementary Fig. 8c).

### $Jt$NorQ$^{WB}$NorD interactions and their functional significance

The two main contact points between NorQ and NorD in the NorQD complex, the finger/H2i interaction and the C-terminus/postS1 loop interaction, contribute differently to the dynamics of the complex. During ATP hydrolysis, AAA+ ATPases move downwards from the top of the spiral arrangement and are normally attached to the substrate via their pore loops as long as they are bound to the nucleotide[8]. The

H2i are located next to the substrate binding loops in clade 7 AAA+ proteins (Fig. 2a). In the structures presented here, they always bind the NorD finger motif in a way that the topmost AAA+ subunits, A, B and C, direct the finger towards the seam. ATP binding to the nucleotide-free seam AAA+ subunit F would cause it to reach upwards and rotate the finger motif anticlockwise by 60 degrees in the hexamer pore, if as a reference frame the subunit location in the hexamer is assumed to be static and seen from the N-terminal side of the complex. In the $Jt$NorQ$^{WB}$D dataset, states 2 and 3 can be distinguished by a rotation of the entire VWA domain (including finger) by 60-degree and 120-degree, respectively, relative to the binding position of the N-terminal domain (Fig. 4a, Supplementary Movie 2). On the other side, the VWA domain is held in place by the interaction of its C-terminus with the postS1 loop, which is bound to the last ADP bound subunit of the AAA+ spiral, subunit E, in all structures of NorQD that show the VWA domain, except for state 4 of $Jt$NorQ$^{WB}$D. Here, the globular VWA domain is rotated 90 degrees anti-clockwise against the finger motif and is located close to the postS1 loop of subunit C (Fig. 4b, c, Supplementary Fig. 12d). The interactions of the C-terminus with the postS1 loop of subunits E and C differ in distance and position of the interaction partners (Fig. 4c). Interestingly, the hydrophobic residues of the postS1 loop (I157/159, L158/160) are oriented towards the centre of the AAA+ pore in ATP bound subunits and rotated away from the pore after nucleotide hydrolysis (Fig. 4e). While ATP bound NorQ has the tendency to point these residues to the channel, because the Sensor 1 (N153) interaction with the nucleotide enforces this geometry of the postS1 loop (Fig. 4d, Supplementary Fig. 14), the loops of ADP bound subunits are usually unstructured (Supplementary Fig. 4a). However, in the presence of the VWA domain I157/L158 are tightly bound to a hydrophobic pocket of this domain (Fig. 4f), suggesting that the VWA domain enforces an ATP-like conformation of the postS1 loop of ADP bound subunits.

## Discussion

The NorQD facilitated iron insertion into $c$NOR, as well as the mechanism of action of the AAA+/VWA domain complex, are still poorly understood mechanistically. Our structural data shows the 'finger-in-pore' interaction mode between NorD and the NorQ H2i region between the AAA+ NorQ and the VWA-partner NorD, which we presume to be general for all AAA+ proteins that interact with VWA domain proteins. In the highly homologous AAA+/VWA domain complex CbbQO (Supplementary Figs. 15 and 16), mutagenesis studies showed the importance of the CbbQ H2i region as well as of the C-terminus of the CbbO for the activase function[15,16] suggesting a very similar interaction mode and action mechanism as the one we suggest for NorQD. The McrBC AAA+ endonuclease complex also shows a very similar finger-in-pore assembly compared to NorQD, however, the partner protein McrC does not contain the VWA domain. Interestingly, the McrB postS1 loops are contacted by the McrC partner protein which was suggested to play a role in ATPase regulation[17]. The human ribosome assembly factor Midasin-1 (Rea1 in yeast) binds its VWA domain to the AAA+ ring centre which is an essential step in pre-60S particle maturation. Cryo-EM structures of Midasin-1 and Rea1[18,19] showed density in the AAA+ central pore that was later identified as the inserted VWA finger[20] (here named MIDAS loop). However, the Midasin-1 inserted finger has not yet been resolved at atomic level.

Our cryo-EM maps obtained after purification under oxidizing conditions show a disulphide bridge in the ATP binding site of the NorQ AAA+ seam subunit. The distorted P-loop is obviously unable to bind fresh ATP, explaining why the ATPase activity of the oxidised complexes are so low. Reversible inhibition of ATPase activity by blocking the binding site has to our knowledge not yet been observed in AAA+ ATPases but appears to be the ideal site for redox enzyme regulation. In some MoxR-like proteins, including CbbQ, the involved cysteines are conserved (Supplementary Fig. 16). As $c$NOR is produced

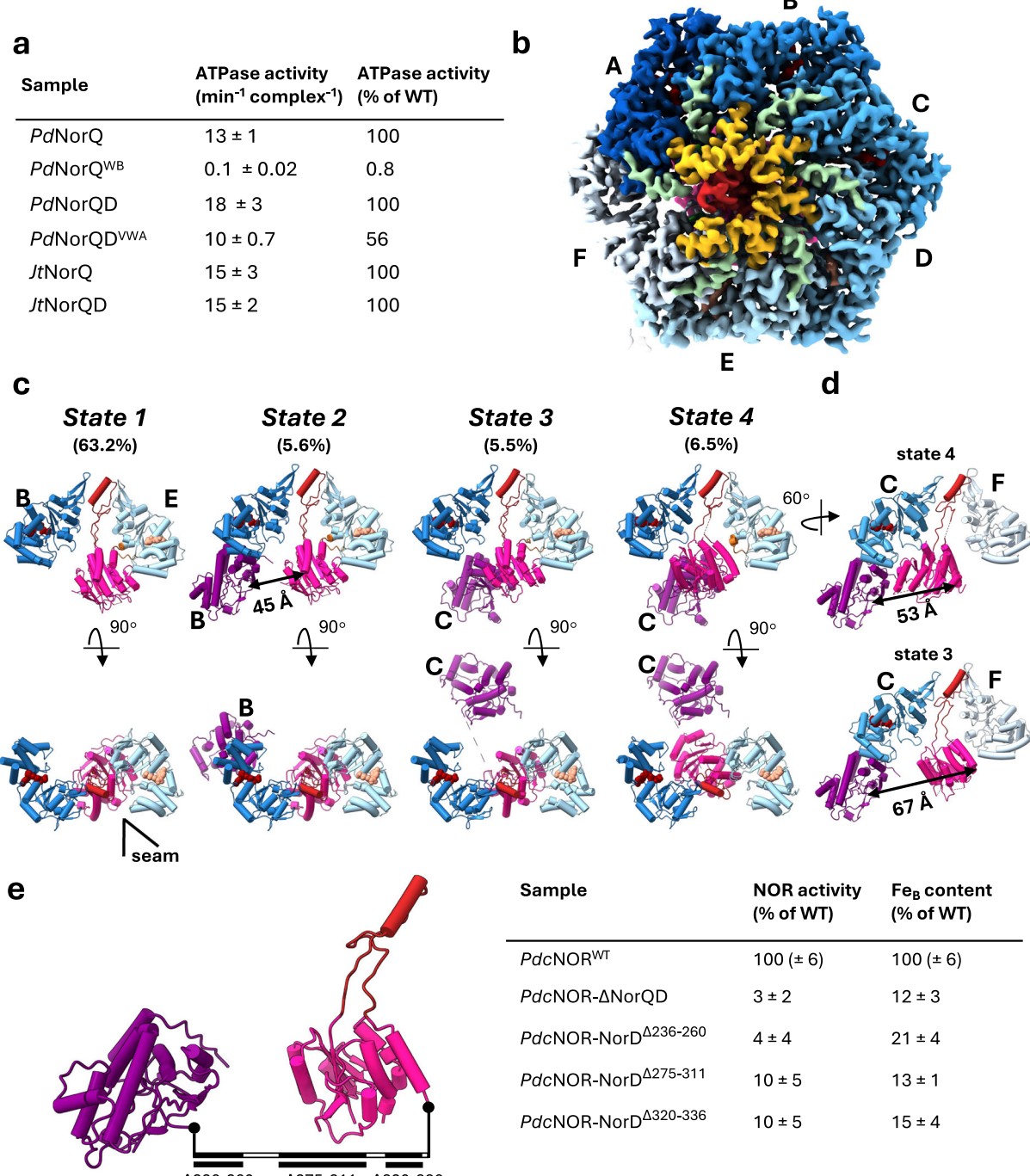

**a**

| Sample | ATPase activity (min$^{-1}$ complex$^{-1}$) | ATPase activity (% of WT) |
|---|---|---|
| *Pd*NorQ | 13 ± 1 | 100 |
| *Pd*NorQ$^{WB}$ | 0.1 ± 0.02 | 0.8 |
| *Pd*NorQD | 18 ± 3 | 100 |
| *Pd*NorQD$^{VWA}$ | 10 ± 0.7 | 56 |
| *Jt*NorQ | 15 ± 3 | 100 |
| *Jt*NorQD | 15 ± 2 | 100 |

**c**

State 1 (63.2%)   State 2 (5.6%)   State 3 (5.5%)   State 4 (6.5%)

**d**

state 4

state 3

**e**

| Sample | NOR activity (% of WT) | Fe$_B$ content (% of WT) |
|---|---|---|
| *Pdc*NOR$^{WT}$ | 100 (± 6) | 100 (± 6) |
| *Pdc*NOR-ΔNorQD | 3 ± 2 | 12 ± 3 |
| *Pdc*NOR-NorD$^{Δ236-260}$ | 4 ± 4 | 21 ± 4 |
| *Pdc*NOR-NorD$^{Δ275-311}$ | 10 ± 5 | 13 ± 1 |
| *Pdc*NOR-NorD$^{Δ320-336}$ | 10 ± 5 | 15 ± 4 |

**Fig. 3 | Structural analysis of *Jt*NorQ$^{WB}$NorD and functional role of the NorD linker. a** ATPase activities of *Pd*NorQ/D and *Jt*NorQ/D complexes. Numbers given are averages with standard deviation (*n* = 3). *Pd* indicates the *P. denitrificans* enzymes, whereas *Jt* indicates the enzymes from *J. thermophila*. WB superscript denotes Walker B mutation, and VWA superscript indicates that the complex was expressed without the NorD N-terminal domain and flexible linker. Source data are provided as a Source Data file. **b** Cryo-EM structure of *Jt*NorQ$^{WB}$NorD at 3.3 Å resolution. The structure is coloured as in Fig. 1. The NorQ protomers are colour-coded in shades of blue according to their position in the spiral, from top (dark) to bottom (light). H2i (helix 2 inserts) are shown in yellow, preS1 (pre-sensor 1) loops in light green, and the tip of the NorD finger is shown in red. **c** Conformational states refined from the *Jt*NorQ$^{WB}$NorD dataset. The proportion of each population in the total dataset is given in %. Major differences are found in the location of the NorD domains. Only models of NorD and NorQ subunits B and E are shown as side view slice (top row) and top view (bottom row). Colour-coding as in Fig. 1 (NorQ coloured in shades of blue according to position in the spiral, NorD$^{NTD}$ purple with the NorQ subunit it binds to in the different states indicated, NorD$^{VWA}$ hot pink and the finger red). **d** Side view slices of state 3 and 4 showing NorD and NorQ subunits C and F. The distances between Cβ atoms of NorD V241 and A393 are indicated as black arrows. **e** NOR activities and non-haem iron content in the *Pd-c*NOR from the NorD linker deletion constructs with a schematic overview (using the *Jt*NorQ/D cartoons) to the left. Numbers given are averages with standard deviation (*n* = 3–4). Source data are provided as a Source Data file.

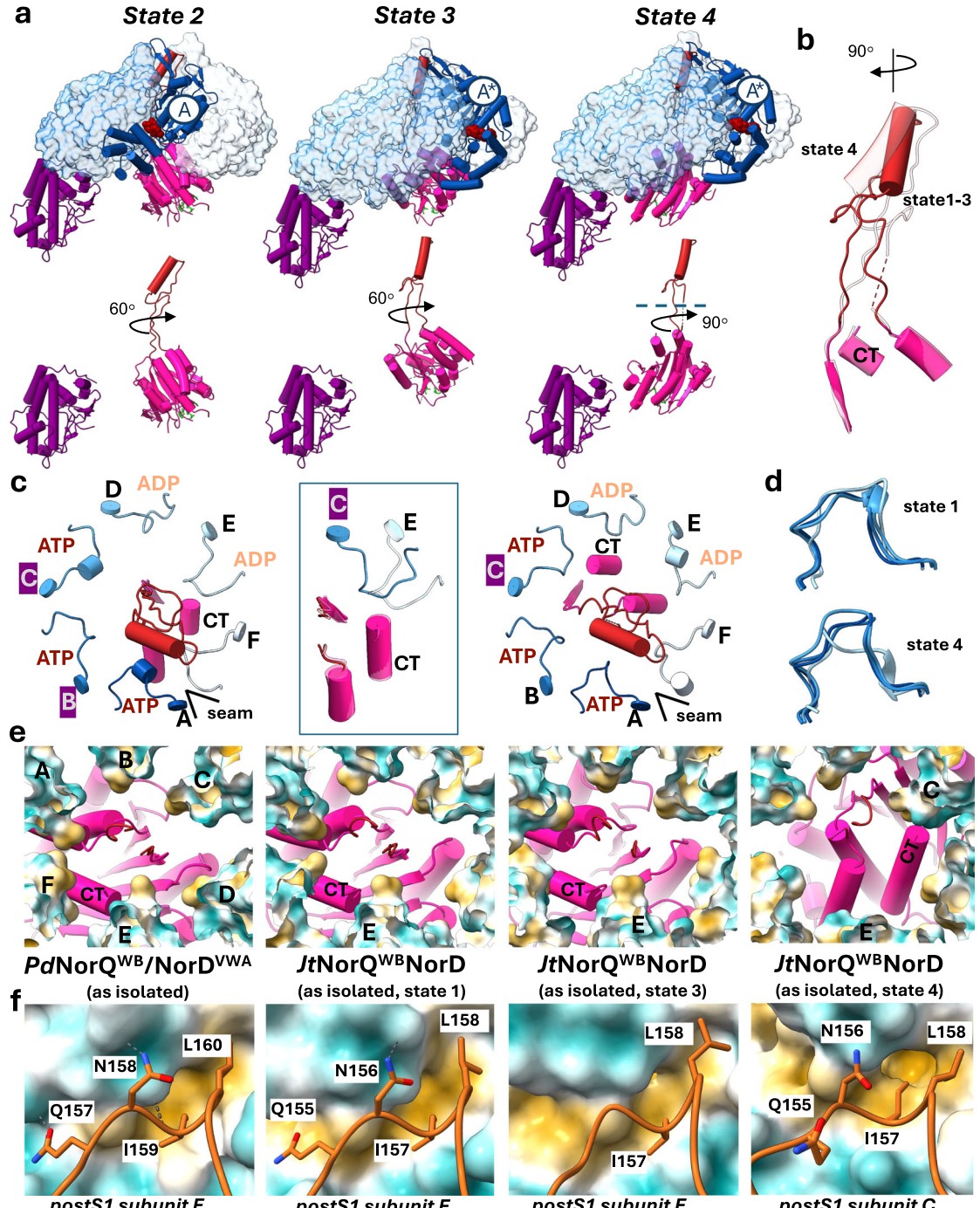

**Fig. 4 | Conformational dynamics in the *Jt*NorQ^WB^NorD dataset. a** Models of states 2–4 are superimposed on the N-terminal domain of NorD (purple) and shown as side views in the top row, with all NorQ subunits represented as transparent surfaces, except for the ATP-loaded seam subunit (A, A*), which is shown as tube helices. A* marks the new seam subunit after ATP binding and reaching up of the bottom seam subunit. The bottom row displays the isolated NorD domains from the above complex in the same viewing direction. ATP molecules shown in red space fill representation. NorD VWA domain is shown as hot pink tube helices with the finger in red. **b** Superposition of the NorD VWA domain, illustrating the rotation between the finger and VWA domains between states 1–3 and state 4. CT denotes the NorD C-terminus. **c** Arrangement of the NorQ postS1 loops (blue) and the NorD VWA domain (pink, finger in red) in states 1–3 (left) and state 4 (right) of *Jt*NorQ^WB^NorD, viewed from the top of the complex. The N-terminal domain of

NorD can bind to either the C or B subunit of NorQ. The orientation of the C-terminus of NorD towards the postS1 loop differs between states 1–3 and state 4 (central panel). **d** Overlay of postS1 loops in NorQ, chains A, B, C and E, of state 1 (top) and state 4 (bottom). The postS1 loop of chain E adopts a conformation as observed in ATP bound subunits when bound to the NorD C-terminus (state 1) but changes conformation when the NorD C-terminus is not bound (state 4).
**e** Hydrophobic interactions in the NorQ pore of *P*dNorQ^WB^/NorD^VWA^ and *Jt*NorQ^WB^/NorD. The postS1 loops of NorQ subunits are shown as surface coloured by hydrophobicity. NorD model is shown as tubes and planks. The NorD C-terminus (CT) and the interacting NorQ protomers are labelled. **f** The postS1 loops of NorD-interacting protomers are shown and individual amino acids labelled. NorD is displayed as surface representation coloured by hydrophobicity (yellow: hydrophobic, cyan: hydrophilic).

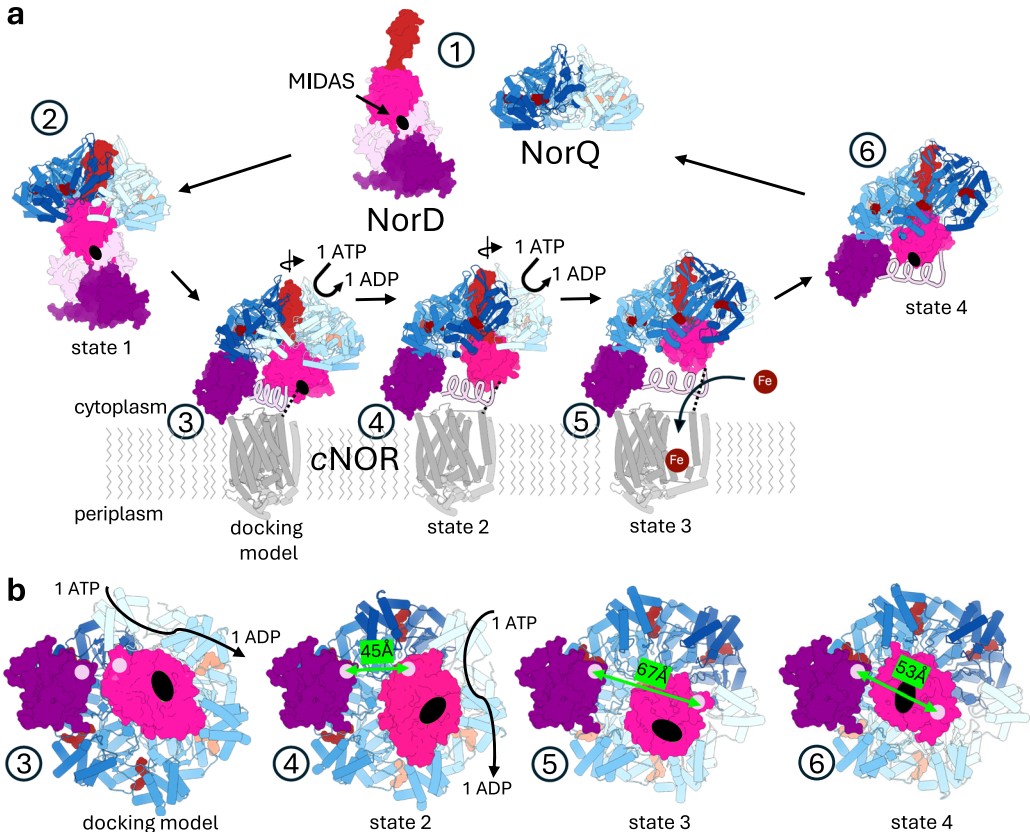

**Fig. 5 | Model of NorQD action. a** Proposed mechanism for NorQD facilitated iron insertion into cNOR. 1. NorQ alone forms a flat, hexameric ring, shown in blue. The AlphaFold 3 model of NorD, with the finger in red, the VWA domain in hot pink, linker region in lilac and the N-terminal domain in purple. 2. NorQ and NorD form a complex, inducing a spiral staircase configuration in NorQ. This conformation resembles state 1 of *Jt*NorQ^WB^NorD. 3. The NorQD complex binds to membrane-bound cNOR (outlined schematically in grey) via NorD, probably involving a) the VWA MIDAS site or linker and two conserved cNOR acidic surface residues and b) at least one additional site on the N-terminal domain and/or the linker. 4. ATP hydrolysis in a hand-over-hand mechanism by NorQ causes rotation of the VWA domain, including the finger domain as seen in state 2 of *Jt*NorQ^WB^NorD. This causes conformational changes in the linker, which is stretched out, hypothetically resulting in conformational changes in cNOR. 5. ATP hydrolysis continues, and the linker is further stretched out, as seen in state 3 of *Jt*NorQ^WB^NorD, which also could lead to further conformational changes in cNOR. This would allow for iron to be inserted into the active site of cNOR. 6. NorQD unbinds upon conformational changes in the linker and VWA domains, state 4 of *Jt*NorQ^WB^NorD. The N-terminal domain of NorD must detach from the ring for a new round of iron insertion. **b** Close-up view onto membrane-facing surface of NorQD complex of steps 3 to 6 as described above with indication of distance between linker attachment points in green. The location of the MIDAS motif is indicated by a black oval.

only during denitrifying conditions, it could be that formation of the specific disulfide bridge is linked to redox regulation in the cell, similar to the Rca AAA+ regulation in plants [21].

Finally, our cryo-EM study, supported by biochemical data, suggests that the AAA+ chaperone NorQ utilizes NorD, which carries the VWA domain, to remodel cNOR through a twisting mechanism (Fig. 5), that is likely to share many features with similar systems. We observe that NorQ alone forms a hexameric ring with low intrinsic ATPase activity. Association with NorD leads to a spiral staircase arrangement in the hexamer and to increased ATPase activity. In the initial stages of contact, the AAA+ complex binds only to the VWA and finger domains of NorD. Conserved residues in the H2i of NorQ mediate the interaction with the finger domain of NorD, while the postS1 loop binds to hydrophobic surface residues at the C-terminus of NorD. The rigid arrangement of the NorD finger and VWA domain can be rotated in the pore of NorQ by each nucleotide binding event and the upwards movement of the newly bound AAA+ subunit in a hand-over-hand mechanism [8,9]. Upon a so far unknown signal, possibly induced by the NorD linker region (lilac in Fig. 5) interacting with target cNOR and thereby unbinding the N-terminal domain from NorD (see Supplementary Fig. 13), the NorD N-terminus binds to the C-terminal domain of the ATP loaded, topmost NorQ subunit A at the spiral seam. Now,

each time a AAA+ subunit reaches the highest position in the helical assembly, the VWA and finger domains of NorD are rotated by ~60° away from the N-terminal domain, stretching the linker domain connecting them. We propose that NorD binds cNOR at two leverage points, one being the N-terminal domain and the other one we suggest to be either its VWA MIDAS motif directly, as supported by previous cNOR mutational and sequence analysis [12,22] or the MIDAS motif-remodelled linker domain. The stretching of the NorD linker would then lead to the ATP dependent opening of cNOR and insertion of an iron ion to the active site through the rotation of the VWA domain. For a new round of metal insertion, the NorQD complex needs to dissociate from cNOR and the N-terminal domain of NorD must detach from NorQ.

## Methods

### Plasmids

Primers (sequences provided in Supplementary Data) were ordered from Eurofins genomics. Genes for expression of *P. denitrificans* NorQ, NorD and the NorD-VWA domain were cloned from the pNOREX vector[23]. *J. thermophila* NorQ and NorD were cloned from the *J. thermophila* strain CC-MHSW-1 (obtained from DMSZ). Genes were amplified using PCR and cloned into pET21a (*P. denitrificans* NorQ) or

pETDuet1 (*P. denitrificans* NorQ, NorD and NorD-VWA domain, *J. thermophila* NorQ and NorD), with the S-tag exchanged for a Strep-tag, for details, see[12]. *P. denitrificans* cNOR was expressed from the pNOREX vector[23], with a C-terminal 6-His tag on NorB[5]. Point mutations, insertions and deletions were done using PCR mutagenesis. *Pd*NorD linker deletions were constructed using primers flanking the deleted region with small overlaps allowing for in vivo recombination of the plasmid, as described in[24]. All cloning was performed in *E. coli* DH5α.

## Protein expression and purification

**His-tagged *Pd*NorQ.** Expression and purification of 6-His-tagged *P. denitrificans* NorQ was done as in[12]. In short, the pET21a vector containing the His-tagged NorQ was transformed to *E. coli* BL21 DE3. Cells were grown at 37 °C, 180 rpm shaking until $OD_{600} = 0.5$. The temperature was then lowered to 30 °C and the cells were left to grow for another 16 h. Harvested cells were resuspended in 50 mM TRIS/HCl, pH 8, 50 mM NaCl, supplemented with one tablet of cOmplete EDTA-free Protease Inhibitor Cocktail (Roche) and DNase I, crushed and ultracentrifuged. The supernatant was mixed with imidazole to a final concentration of 20 mM and applied on a 5 mL HisTrap (Cytiva) column using an ÄKTA purification system (GE Healthcare). Protein was eluted in 20 mM TRIS/HCl, pH 8, 300 mM NaCl, 10% (v/v) glycerol with an imidazole gradient (50 mM-500 mM). Eluted protein was dialysed against 10 mM TRIS/HCl, pH 8, 150 mM NaCl, and 10% (v/v) glycerol overnight. Protein concentration was determined using the Bradford assay.

**Double-tagged NorQD and NorQD^VWA.** pETDuet1-strep constructs with NorQ and NorD or NorD-VWA domain genes were transformed to *E. coli* BL21 DE3, grown at 37 °C, 180 rpm, in LB medium to OD 0.5. The temperature was then lowered to 20 °C. Cells were harvested after 16 h and resuspended in 50 mM TRIS/HCl, pH 8, 50 mM NaCl, supplemented with one tablet of cOmplete, EDTA-free Protease Inhibitor Cocktail (Roche) and DNase I, with or without the addition of 10 mM DTT and 2 mM ADP, followed by cell disruption at 22 kpsi (Emulsiflex C3 Homogenizer) and ultra-centrifugation. The supernatant was applied to a 7 mL Strep-Tactin column and washed with buffer (20 mM TRIS/HCl, pH 8, 300 mM NaCl, 10% (v/v) glycerol, with or without 10 mM DTT and 2 mM ADP). Protein was eluted with the same buffer, supplemented with 10 mM desthiobiotin. The eluted protein was mixed with 1 mL Ni-NTA resin and imidazole was added to a concentration of 20 mM, followed by 1 h rotating incubation. The resin was washed with buffer containing 50 mM imidazole and protein was eluted with 500 mM imidazole. The eluted sample was dialysed against dialysis buffer (10 mM TRIS, pH 8, 150 mM NaCl, and 10% (v/v) glycerol, with or without 10 mM DTT) overnight. Purified protein was analysed on SDS-page gel (Supplementary Fig. 8a–c) and concentration was determined using the Bradford assay.

**Strep-tagged *Pd*NorQ and *Jt*NorQ.** Expression of Strep-tagged NorQ without NorD or the VWA domain from *P. denitrificans* and *J. thermophila* was done from the pETDuet1-strep constructs, with the first MCS left empty. Expression and purification were done similarly as for the NorQD/VWA complexes, with some exceptions: cells were grown at 30 °C, all buffers were without DTT and ADP, and the Ni-NTA purification and dialysis was not performed.

**His-tagged cNOR.** Expression and purification of 6-His-tagged *P. denitrificans* cNOR was done using the pNOREX plasmid with a C-terminal 6-His tag on NorB[5]. Wild-type pNOREX, pNOREX with the NorD finger deleted (residues 403 to 448 exchanged for GGSIDGGS) and pNOREX with the NorD linker deletions, was transformed to *E. coli* JM109, also holding the pEC86 vector, enabling haem c synthesis in *E. coli*. The cells were grown in TB media at 37 °C, 180 rpm, until $OD_{600}$

reached 0.5. Protein production was induced with 1 mM IPTG and the temperature was lowered, first to 30 °C for 5 h, then to 18 °C and the shaking was turned off. After an additional 5 h, the temperature was increased to 30 °C and shaking 180 rpm until harvesting. Harvested cells were resuspended in 100 mM TRIS/HCl, pH 7.6, 50 mM KCl, supplemented with DNase I and 1 tablet of cOmplete EDTA-free protease inhibitor cocktail (Roche), crushed at 22 kpsi and ultra-centrifuged. Pelleted membranes were solubilized in 1% n-dodecyl-β-D-maltoside (DDM) for 1 h and ultracentrifuged again. The solubilized membranes were mixed with 5 mL Ni-NTA resin and 20 mM imidazole was added. The column was washed with buffer (50 mM imidazole, 50 mM TRIS/HCl, pH 7.6, 200 mM KCl, 0.05% DDM) and protein was eluted in the same buffer with 250 mM imidazole. Eluted protein was concentrated, and concentration was estimated using $\varepsilon_{410, ox} = 311 \, cm^{-1} \, mM^{-1}$.

## Enzymatic assays

**ATPase Activity measurements.** ATP hydrolysis activity measurements were done using the malachite green end-point assay[5,25]. The protein was diluted to 25–60 µg/mL in 20 mM TRIS/HCl, pH 8, 300 mM NaCl, 10% (v/v) glycerol, 15 mM $MgCl_2$, with or without 2 mM DTT. To start the reaction, ATP was added to a final concentration of 2 mM. At each time point (1–61 min after start), 27.3 µL of the reaction was mixed with 109.1 µL malachite green colour reagent (0.034% malachite green oxalate, 1% ammonium molybdate tetrahydrate, 0.04% Triton-X) followed by vortexing. After 1 min, 13.64 µL 34% citrate solution was added, and the sample was again vortexed. $A_{620}$ was measured 5 minutes later, and the concentration of $P_i$ was calculated using a standard curve.

**cNOR activity measurements.** NO-reduction activity was measured using an NO-sensitive electrode and activity values calculated from maximum slopes[12]. cNOR was diluted to 12.5–25 nM in 50 mM HEPES, pH 7.5, 50 mM KCl, 0.05% DDM, with 30 mM glucose and 10 U/mL catalase. The chamber was closed, and glucose oxidase was added to a concentration of 1–3 U/mL to deplete the chamber of oxygen. NO was added in steps of 10 µM up to 30–50 µM. Electron donors (0.5 mM TMPD, 3 mM ascorbate, 10 µM bovine cytochrome c) were added and the NO consumption was followed over time.

## Non-haem iron

The non-haem iron content in cNOR was measured using the ferene method[26,27]. The cNOR protein was diluted to 7 µM in 60 µL buffer (50 mM HEPES, pH 7.5, 50 mM KCl, 0.05% DDM). Protein was precipitated by addition of 6 µL HCl (37%) and incubated at room temperature for 5 min. Upon centrifugation, the supernatant was mixed with 100 µL of 3 M sodium acetate and 10 µL 1 M ascorbic acid (pH 6) and the absorbance at 593 nm was recorded. Ferene (3 mM, 10 µL) was added, and the sample was incubated in the dark for 5 min. The $A_{593}$ was measured again and the difference in $A_{593}$ before and after addition of ferene was used to estimate the non-haem iron content using $\varepsilon_{593} = 35.5 \, mM^{-1} \, cm^{-1}$ for the ferene-iron complex [26].

## Bioinformatic analysis

Sequence conservation of NorQ and NorD was analysed by alignment of 20 sequences for each protein using Clustal Omega[28]. The Uniprot accession numbers for the used sequences are: Q51664; A0A1H5TIV7; Q5LL14; B8EHY5; Q16A05; Q3JI36; A0A1G5P588; V6F0A1; A0A1H8I2H0; Q7CUT7; Q51481; A4VQB0; E4U3I4; E0UUV4; Q0A6S6; Q82TA2; A6X784; B1Y7Z3; H7C804 and C6WVL4 for NorQ and Q51665; A0A1H5TKL0; Q5LL15; B8EHY6; Q16A06; Q3JI37; A0A1G5P5T9; V6F0E0; A0A1H8I2C8; A9CGJ7; Q51484; A4VQ97; E4U3I3; E0UUV3; Q0A6S5; Q82TA1; A6X785; B1Y7Z1; Q89QB3 and C6WVL5 for NorD. The sequence conservation of selected regions was illustrated using WebLogo [29].

## Cryo-EM sample preparation and data acquisition

In general, 3 μL protein sample (0.5–1 mg/mL) was applied to freshly glow-discharged C-flat Cu 2/2 (300 and 400 mesh) grids and then plunge-frozen on a Vitrobot Mark IV at 4 °C and 100% humidity. The images were collected on a Thermo Fisher Titan Krios equipped with a Gatan K3 BioQuantum detector using the Thermo Scientific EPU 3 software. Detailed information for each data acquisition session can be found in Tables 1–3.

For the $Pd$NorQ$^{WB}$NorD$^{VWA}$ samples in presence of ATP, 4 mM ATP and 10 mM MgCl$_2$ was added immediately prior to grid preparation.

## Cryo-EM image processing

Image processing was performed in the software packages RELION 4[30,31] and CryoSPARC[32]. In brief, movie frames were aligned using MotionCor2[33] and contrast transfer functions were calculated with CTFFIND4[34]. Particles were picked using Gaussian blob template-free auto picking. The resolution reported after 3D refinements were calculated using gold standard Fourier Shell Correlation (FSC) curves with the FSC = 0.143 criterion for mask-corrected FSC curves. For final 3D refinements dynamic auto-tightened masks were used.

For $Pd$NorQ$^{WB}$ ~ 3 million particles were picked from 11,678 micrographs and cleaned in several rounds of 2D classifications in CryoSPARC (Supplementary Fig. 1). However, the 2D classes showed severe preferred orientation. We therefore collected a tilted dataset (30°) with 2658 micrographs yielding ~1.4 million particles. After several rounds of 2D classification and 2D class balancing, good particles from both datasets were combined for downstream processing. After ab initio modelling, the particles were sorted by heterogeneous refinement (4 classes) and the best class with 113,836 particles was refined to an overall resolution of 3.7 Å.

For $Pd$NorQ$^{WB}$D$^{VWA}$(oxidized) ~3.3 million particles were picked and extracted from a total 10,893 micrographs in RELION (Supplementary Fig. 3). After multiple rounds of 2D classification and ab initio modelling the cleaned particles were subjected to 3D classification with 3 classes. One class with poor resolution was discarded, and the remaining particles were refined to an overall resolution of 3.0 Å. However, the local resolution of the NorD VWA domain was relatively low. The particles stack was transferred to CryoSPARC for further analysis. A subsequent 3D classification yielded 10 classes with subtle differences in the position of the NorD VWA domain and no variation within the NorQ hexameric arrangement revealed a slight tilting motion of the VWA. The 3D class that yielded the highest local resolution and connectivity in the NorD VWA region after homogeneous refinement (overall 3.2 Å) was used for molecular model building. The tilting motion of the VWA was visualized using multi-body refinement in Relion (Supplementary Movie 1).

For the $Pd$NorQ$^{WB}$D$^{VWA}$ sample in presence of ATP we extracted ~4 million particles from 16,559 micrographs in CryoSPARC (Supplementary Fig. 7). After multiple rounds of 2D classification the cleaned particles were further sorted using ab initio modelling and heterogeneous refinement with 3 classes. Each of the three subsets represents a distinct oligomeric state of the complex and attempts to further separate the particles did not yield additional states or maps of higher resolution. Homogeneous refinement of the 3 subsets yielded an overall resolution of ~3 Å.

For $Pd$NorQ$^{WB}$D$^{VWA}$ (reduced) in presence of DTT ~1.8 million particles were picked from 7273 micrographs (Supplementary Fig. 5). Relatively few particles (100,884) remained after cleaning in multiple rounds of 2D classification likely due to the dataset suffering from relatively thick ice. The particles were further cleaned by 3D classification using 4 classes which yielded only one class suitable for further processing. The particles of the best class were exported as a particle stack and used for homogeneous refinement in CryoSPARC yielding a map of 3.8 Å.

For $Jt$NorQ$^{WB}$D ~ 3.4 million particles were picked from a total of 11,373 micrographs in RELION (Supplementary Fig. 9). After multiple runs of 2D classification and subsequent ab initio modelling, the particles were sorted by 3D classification with yielding two poorly resolved classes and two good classes. The particles of both good classes were transferred to CryoSPARC for subsequent processing. The 3D models produced to this point showed weak density for the NorD NTD bound to the NorQ hexamer. The imported particles were sorted by 3D classification using 20 classes using a focus mask including both, the NorD VWA and NTD. 13 classes (~260,000 particles) showed an identical assembly compared to $Pd$NorQ$^{WB}$D$^{VWA}$ with no or only faint density for the NorD NTD (State 1). 5 classes (~104,000 particles) showed relatively low resolution of the NorD VWA, and these particles were not used for molecular model building. One class (~23,000 particles) contained particles similar to State 1 but in addition showed clear density for the NorD NTD (State 3). A final class (~27,000 particles) showed clear density for the NorD NTD, but the VWA domain was rotated compared to State 1 and 3 (State 4). The density of the NorQ hexameric assembly was virtually identical in all classes on the scale of considering the asymmetric placement of NorQ protomers. Up to this point the NorD NTD was observed bound to only one specific NorQ protomer in the asymmetric NorQ spiral. We tested possible alternative binding positions of the NTD at the NorQ small subdomain by extensive 3D classifications with spherical masks at the expected binding sites. The six parallel 3D classification runs (one run for each NorQ protomer with 20 classes) yielded one additional binding position of the NorD NTD in a 3D class with ~23,000 particles (State 2). Homogeneous refinement of each particle subset yielded an overall resolution of 3.3 Å (State 1), 3.8 Å (State 2), 3.9 Å (State 3) and 3.9 Å (State 4).

## Molecular model building and analysis

Individual domains of NorQ and NorD models generated with Alpha-Fold 3[35] were fitted as rigid body into the density using UCSF ChimeraX[36]. The models were optimized manually in COOT[37] at the level of individual amino acids using ProSMART secondary structure restrains[38] derived from the AlphaFold 3 models. The models were then refined automatically using phenix.real_space_refine in Phenix[39]. Iterative model adjustment in COOT and subsequent refinement in Phenix were performed until the models were completed with a final real space refinement in Phenix. UCSF ChimeraX was used to analyse the models and to generate the model figures. The atomic models of $Jt$NorQ$^{WB}$D state 2–4 were refined and deposited without the NorD NTD as the local resolution and connectivity of the maps was suboptimal for model building in this part. For the figures we fitted the AlphaFold 3 model of the NorD NTD as a rigid body and the corresponding composite models are available via the Zenodo repository (see Data availability statement). Surface representations that show hydrophobic potentials are displayed according to ChimeraX standard colour code: dark goldenrod for the most hydrophobic potentials through white to dark cyan for the most hydrophilic potential. Interaction parameters for atom pairs correspond to VDW overlap of ≥ - 0.4 Å for all contacts and standard geometric parameters for hydrogen bonds.

## Reporting summary

Further information on research design is available in the Nature Portfolio Reporting Summary linked to this article.

# Data availability

The cryo-EM maps have been deposited in the Electron Microscopy Data Bank (EMDB) under accession codes EMD-53154 ($Pd$NorQ$^{WB}$); EMD-53156 ($Pd$NorQ$^{WB}$D$^{VWA}$); EMD-53158 ($Pd$NorQ$^{WB}$D$^{VWA}$ reduced); EMD-53157 ($Pd$NorQ$^{WB}$D$^{VWA}$ATP subset 1); EMD-53159 ($Pd$NorQ$^{WB}$D$^{VWA}$ATP subset 2); EMD-53160 ($Jt$NorQ$^{WB}$D state 1); EMD-

53161 (*Jt*NorQ^WBD state 2); EMD-53162 (*Jt*NorQ^WBD state 3) and EMD-53163 (*Jt*NorQ^WBD state 4). The atomic coordinates have been deposited in the Protein Data Bank (PDB) under accession codes 9QH4 (*Pd*NorQ^WB); 9QH6 (*Pd*NorQ^WBD^VWA); 9QH8 (*Pd*NorQ^WBD^VWA reduced); 9QH7 (*Pd*NorQ^WBD^VWAATP subset 1); 9QH9 (*Pd*NorQ^WBD^VWAATP subset 2); 9QHD (*Jt*NorQ^WBD state 1); 9QHE (*Jt*NorQ^WBD state 2); 9QHF (*Jt*NorQ^WBD state 3) and 9QHG (*Jt*NorQ^WBD state 4). The atomic models of *Jt*NorQ^WBD state 2–4 with a rigid body fit of the NorD NTD were deposited to the Zenodo repository [https://doi.org/10.5281/zenodo.17936648][40]. Source data are provided with this paper.

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

## Acknowledgements

We thank Dr. Maximilian Voit for IT support. The work was funded by the Deutsche Forschungsgemeinschaft (DFG, German Research

Foundation) project numbers 493617395 (to MK) and 406260942 (TEM to PW), and by the Swedish Research Council VR grant nr 2019-04124 (to PÄ). The cryo-EM data was collected at the Cryo-EM Swedish National Facility funded by the Knut and Alice Wallenberg, Family Erling Persson and Kempe Foundations, SciLifeLab, Stockholm University.

## Author contributions

Design of study: M.K., S.A., P.Ä. and P.W.; Data acquisition: M.K., S.A., F.K.; Data analysis: M.K., S.A., F.K., M.C., P.Ä. and P.W.; Writing, original draft: M.K., S.A. and P.W.; Writing, review and editing: M.K., S.A., F.K., M.C., P.Ä. and P.W.; Funding acquisition: M.K., P.Ä. and P.W.

## Funding

## Competing interests

The authors declare no competing interests.
