## [Transparent Peer Review file · Nature Communications]

NorQD AAA+ complex drives metal insertion by a twisting mechanism

Corresponding Author: Professor Petra Wendler

Version 0:

Reviewer comments:

Reviewer #1

(Remarks to the Author)

In this manuscript, Kahle et al. determine the structure of the MoxR AAA+ ATPase NorQ and its partner VWA protein NorD using cryoEM. NorQD have been shown to facilitate iron insertion into cNOR (cytochrome C -dependent nitric oxide reductase); cNOR is involved in nitrate respiration. The authors show that a finger protruding from the VWA domain of NorD inserts into the central pore of the NorQ hexamer. The authors also show that the the C-terminus of NorD interacts with a post sensor 1 loop in NorQ. The authors obtain the cryoEM structure of NorQD from *Paracoccus denitrificans* and from *Jhaorihella thermophila* with the latter being at higher resolution. The structures are interesting. The structural data are further verified using mutational analysis, interaction studies, and ATPase assays. The authors propose a mechanism whereby NorQD 'twists' and stretches the NorD linker to enable metal insertion into its target cNOR.

While the structures are interesting, there is no evidence for the proposed action of NorQD on cNOR. In other words, the authors do not carry out experiments to demonstrate the proposed "twisting" mechanism. In the absence of such data, the manuscript is basically showing a series of structures of NorQD.

1. The abstract is poorly written with no explanation of the function of NorQD. The abstract concludes with a proposed function of NorQD on cNOR, which is not experimentally addressed in the manuscript.
2. Since the JtNorQD is obtained at higher resolution than that of PdNorQD, it might be better to start the manuscript with describing the JtNorQD structure.
3. In line 74, the authors should provide an explanation as to why the Walker B mutant was used for cryo-EM throughout the paper. They did say the mutant has much lower ATP hydrolysis, but they did not say why they did not use wild type for cryoEM. They could have used ADP.
4. Line 81/82: 'Bottom' and 'top' are relative terms. When describing the subunits, it might be better to use the labels presented in the figures, i.e. subunits A-F. Similarly, throughout, when describing the seam subunits, refer to them by their label as well, to increase clarity.
5. Line 109: Why was excluding ATP useful in obtaining this structure? Why would it have affected the resolution?
6. Line 109: The structure only includes the VWA domain of PdNorD; the other domains/regions might play a role in the interaction as well, so it would be good to introduce the complex with the understanding of this limitation.
7. On line 118, the 'finger' was suggested to aid in nucleotide exchange, but activity assays showed greater ATPase activity without this finger. These data conflict with the earlier statement. How may they be reconciled?
8. Does the disulfide bridge play any biological role, and has it been observed in other homologs?
9. Supplementary table 4 indicates assays were conducted in DTT, but that was not sufficient for full activity as in supplementary table 6. Therefore, all the other constructs and mutants also need to be purified with DTT.

10. In Supplemental Figure 4, the subsets 2 and 3 of have low cFAR values, which can indicate some extent of orientation bias. Perhaps the authors can comment on how confident they are regarding the biological relevance/the conclusions made on the two subsets. This puts into questions some of the mechanisms they are proposing.
11. Line 163, when reporting RMSD values, authors should mention if it is using all atoms of the relevant amino acids, or all heavy atoms, or just alpha carbons etc.
12. Explain the dotted circle in the panels of Figure 1b.
13. The discussion could more explicitly acknowledge the limitations of the current data, especially the lower resolution in the VWA domain region and the lack of direct visualization of remodeling, and should consider alternative models.
14. Figure legends could be more detailed, especially regarding the number of replicates and statistical tests for the biochemical assays presented.
15. Although there is an overall schematic (Figure 5), it could benefit from more detail or a zoomed-in feature highlighting the proposed mechanism.

Reviewer #2

(Remarks to the Author)

This manuscript reports cryo-EM structures of the MoxR AAA+ ATPase NorQ in complex with its VWA domain partner NorD. NorQ and NorD are essential for insertion of an Fe into the membrane bound cytochrome c-dependent NOR (cNOR). Coupled with ATPase measurements, the authors propose a structural model for how the NorQD complex induces structural changes in cNOR. The structures are compelling and shed light on an essential biological function. The manuscript should be published.

Reviewer #3

(Remarks to the Author)

Kahle and colleagues present multiple structures of the AAA+ ATPase, NorQ, alone or with its VWA domain partner, NorD. NorQB structures were determined of both *Paracoccus denitrificans* and *Jhaorihella thermophila* homologs, and in oxidizing or reducing conditions. The JtNorQD cryo-EM data analysis resulted in four unique states, including one with only the VWA domain of NorD shown, and three states with unique NTD conformations of NorD. The authors conclude that these states represent the rotation of the VWA and NTD, leading to ATP dependent opening of cNOR and insertion of an iron ion.

The cryo-EM and image analysis was done to a high standard and well documented.

Major points:

1. The order in which these structures are presented seems backwards of a logical flow of information.

It is unclear why the authors do not include a reducing agent in their protein purifications, experiments, and cryo-EM samples in the first place. The way the manuscript is written, it seems like an oversight that is then corrected by redoing the experiment. If there is a biological reasoning behind this, as is alluded to on Lines 258-260, please introduce it earlier. You could also start with the structures containing DTT, and then introduce the disulfide bridge found in the oxidized structure after. It seems that with an RMSD of ~1Å between them, your conclusions in the results section won't change much with this reorganization, and you can say that the under oxidizing conditions the ATPase activity is reduced, likely due to this disulfide forming.

I feel similarly about the current order of starting with describing the Pd NorQD structures then transitioning to Jt NorQD. It seems that state 1 of the JtNorQD structures is the same as PdNorQD (RMSD 0.687Å). The interesting results are coming from the conformational switching between the NorD VWA/NTDs. It seems that the reason for this ordering is because there is significant anisotropy in the JtNorQD structures, limiting ability to solve the structures from those maps alone. I think there is nothing wrong with this, but the authors could do a better job creating a logical flow from one set of structures to the next.

2. The significance of the spiral staircase change in NorQD structures is unclear.

Starting with the title on line 90, the authors describe the staircase of NorQ as having a "steeper spiral staircase" when bound to NorD, as compared to the structure without NorD. In Line 101 authors say the ATP "complexes display a significantly steeper spiral staircase and stronger seam than seen in the 'as isolated' PdNorQWB complex". What do you mean by 'as isolated'? Is this the previous structure? Be clear here, as you use this wording a few more times.

Define 'significance' in the change in the staircase? Can it be described in a distance change? Unfortunately, the figures do not provide enough information about this change. Rather than showing the maps with slightly altered protomer E in Supplementary Figure 2, you should have the atomic structures of subunits of F, E and D overlaid so we can see their movement down relative to each other. Perhaps it would be more appropriate to show different subunits- it depends which subunit to which you are aligning the structures.

In Lines 102-105 the authors also describe the loss of subunits in the ring. The maps in Supplementary Figure 2 look like there is missing density in part of the subunits E and F, which likely implies flexibility and heterogeneity of particles in your data processing. Knowing this, it is hard to compare the “movement” of these subunits, as some may appear lower in the staircase compared to their counterparts that are entirely lacking that part of the subunit.

3. The order of progression between states in JtNorQD is confusing and hard to understand from the figures and movies.

When you are comparing an ensemble of models of a homohexameric complex it can be unclear the order in which they progress in the mechanism, or how to assign the ‘top’ subunit for alignment. When watching the movies, it seems that the VWA domain is rotating a ton, with little movement of the ATPase. I wonder what the progression would look like if you instead aligned the structures based on the VWA domain to see how the subunits of NorQ moved relative to it. NorQ is the ATPase and is known to go through large conformational switching while processing substrates. It seems more likely it would be moving, and the rotation of VWA would be in response to this.

Either way, the authors need to find a better way to demonstrate these conformational changes in their figures and movies. The current angles shown in Figure 3 are confusing, and you should find a better angle than a 90-degree rotation so we can understand that swing of the VWA domain.

4. Proposed mechanism includes the substrate, cNOR, but the manuscript does not provide direct evidence for how NorQD interacts with this substrate.

The structures presented in this manuscript lack the membrane-bound substrate, cNOR. In Figure 5 they include cNOR and provide a hypothesis of how these structures would interact with that substrate. Is there previous data to suggest that NorQ does not translocate a part of cNOR into the central channel? If so, would both NorD and cNOR be accommodated in the channel? What is the evidence that NorQD would sit atop of cNOR in this way? The authors need to provide better justification for this hypothesis, which currently has no evidence from the data presented.

Minor points:

Some of the figures could use some adjustments for clarity.

1. In Figure 2, please show both residues involved in the bonds, including in the backbone. The way the structure is being displayed makes it hard to see the atoms involved in the interaction.
2. In the Figure 3 legend you need to include a description of what colors associate with which domain. It is not clear here which domain is VWA or NTD of NorD.
3. In Figure 5 you should add arrows between states. Would be helpful to have the structure/state on the figure itself, rather than just listed in the figure legend.

The methods section was written rather poorly, including what appear to be typos, informal writing style and incorrect shorthand. Please edit the methods section to have a similar writing style as the rest of the manuscript and correct the mistakes.

1. Line 638 – “BL21 DE. Cells” – do you mean BL21 DE3?
2. Line 640 – When describing protease inhibitor tablets it was called a “pill” on Line 640 and 652 but called a “tablet” to describe it on line 681. Be consistent.
3. Line 643- Authors claim to use a “HiTrap” column for purification but use an imidazole gradient. Do you mean a “HisTrap”?
4. Line 737 – To describe the total number of particles, the shorthand “3 mio.” was used. Do you mean million? Either write it out or find a more appropriate shorthand.
5. An example of informal writing can be found on Line 772 – “two trash classes”. What do you mean by trash classes? Find a better way to describe your process for eliminating bad classes.

Version 1:

Reviewer comments:

Reviewer #1

(Remarks to the Author)

The authors made some effort to make the suggested changes. They do additional experiments with truncations of the linker domain that essentially show inactive protein, while still being able to form the complex. This is to try to address the lack of experimental data to support their model. They explained away other issues I raised and they made all the clarity changes I wanted for the text, but not really to the model figure. For that figure, they just added directional arrows and a couple of labels.

Reviewer #2

(Remarks to the Author)

The authors have adequately addressed the reviewer comments.

Reviewer #3

(Remarks to the Author)

Reviewer Major Point (from original review):

2. The significance of the spiral staircase change in NorQD structures is unclear

Starting with the title on line 90, the authors describe the staircase of NorQ as having a “steeper spiral staircase” when bound to NorD, as compared to the structure without NorD. In Line 101 authors say the ATP “complexes display a significantly steeper spiral staircase and stronger seam than seen in the ‘as isolated’ PdNorQWB complex”. What do you mean by ‘as isolated’? Is this the previous structure? Be clear here, as you use this wording a few more times.

Define ‘significance’ in the change in the staircase? Can it be described in a distance change? Unfortunately, the figures do not provide enough information about this change. Rather than showing the maps with slightly altered protomer E in Supplementary Figure 2, you should have the atomic structures of subunits of F, E and D overlaid so we can see their movement down relative to each other. Perhaps it would be more appropriate to show different subunits- it depends which subunit to which you are aligning the structures.

Author Response: We have now added guidelines and distance marks to Supplementary figures S2a,S2c and S6c to highlight the change in the spiral staircase arrangements. In our hands, the change is best seen, when comparing the maps. An overlay of the atomic structures, even when only showing Ca traces, is very overwhelming. In AAA+ proteins, the helical shift between subunits is mostly dependent on the nucleotide bound to the subunits. Hence, the shift in the ATP bound subunits A-C is similar, but differs for the ADP bound subunits D-F as the reviewer points out correctly. It therefore makes most sense to align all spirals on the topmost subunit in the spiral, which is subunit A. And this is what we have done. We hope that the changes we introduced could clarify this point.

Reviewer Response: Authors still have not defined what they deem as a ‘significant’ change in the staircase. Please include comments in text about the distance change.

I disagree that having an overlay of the atomic structures would be overwhelming. You do it in Supplementary figure S2e. You could make a similar figure by showing the atomic structure of subunit E as an overlay of the “as isolated” with each of the ATP structures, as they are currently aligned in figure S2 a and b.

Reviewer Major Point (from original review):

3. The order of progression between states in JtNorQD is confusing and hard to understand from the figures and movies.

When you are comparing an ensemble of models of a homohexameric complex it can be unclear the order in which they progress in the mechanism, or how to assign the ‘top’ subunit for alignment. When watching the movies, it seems that the VWA domain is rotating a ton, with little movement of the ATPase. I wonder what the progression would look like if you instead aligned the structures based on the VWA domain to see how the subunits of NorQ moved relative to it. NorQ is the ATPase and is known to go through large conformational switching while processing substrates. It seems more likely it would be moving, and the rotation of VWA would be in response to this.

Either way, the authors need to find a better way to demonstrate these conformational changes in their figures and movies. The current angles shown in Figure 3 are confusing, and you should find a better angle than a 90-degree rotation so we can understand that swing of the VWA domain.

Author’s Response: In our initial movie, we aligned the structures to the finger motif of the VWA domain and showed how all other domains (including the AAA+ subunits) moved relative to it. Now, we have revised the movie comparing all JtNorQD states such that we kept the subunit bound to the NorD N-terminal domain static so that all other domains move relative to it. We have also labeled the linker attachment points in NorD for more clarity. As can be seen very well in the movie, there is no movement between state 3 and state 4 apart from the flipping back of the NorD VWA domain. All other structural features are identical. Dynamic movements, however, can be depicted in various ways relative to each other. One could argue that in the NorQD system, the NorD N-terminus is bound to a target on the membrane as well as the NorQ ring and the rest of the system moves relative to this attachment point. Hence, we kept the N-terminal domain static showing this scenario of events. To us it is clear how to assign the top subunit for alignment. The top subunit will always be the first ATP bound

AAA+ domain in the spiral, next to the linker domain of NorD.

Reviewer response: Authors say that it is clear to assign the top subunit as the first ATP bound AAA+ domain in the spiral. I agree, and I can see that is how they have assigned it in Supplementary Figure 11. But, since the movie is aligned to the NTD, this is changing how the other subunits are assigned.

In the Supplementary figure 11, the NTD is bound to subunit B in state 2, and subunit C in state 3. Now that you have aligned them to the NTD in the movie, in the transition between state 2 and state 3 we see the NTD bound to subunit B in both, and therefore now we see subunit F go from being a seam subunit to the new top of the spiral staircase in this transition, with subunits D, E and F all moving in an upward direction as a rigid body. Have you tried aligning to the NTD and morphing from state 3 to state 2? It seems like you might have it backwards, based on what we know of AAA+ motor translocation mechanisms.

Evidence that the motor moves in a hand-over-hand mechanism can be found in numerous studies. It is particularly easy to see in AAA+ motor structures of the 26S proteasome, which has a heterohexameric AAA+ that can be aligned to the 20S core peptidase; therefore, movement of each subunit can be followed through the ATP hydrolysis cycle. In Dong et al. Nature 2019 and de la Peña et al. Science 2018, they saw consecutive ATP hydrolysis and translocation states in which the ATP-bound subunits move downwards as a rigid body and the bottom subunit releases substrate and rebinds at the top of the spiral. Assignments of 'top' and seam subunits is done the same way as the authors described in this study.

I agree with authors that you need to pick something to align to, and in your study it makes sense to align all the models to the NTD, since that is likely bound to a membrane. This would require more self-consistency in the manuscript. You should align your models to the NTD in Supplementary Figure 11a and b. You should also try morphing from state 3 to state 2 and see if that direction of movement is more consistent with previous studies, in which rigid body movement of the ATP-bound subunits is downward, and the seam unit will move up to become the new 'top' subunit. This could show an interesting result of how the movement of the NTD and VWA is related to a translocation event.

All other major and minor comments were addressed by authors.

Version 2:

Reviewer comments:

Reviewer #1

(Remarks to the Author)
I had no further comments.

Reviewer #3

(Remarks to the Author)
The authors have adequately addressed the reviewer comments.

We thank all reviewers for their constructive criticism of our manuscript, which we believe has strengthened the manuscript and made it easier to understand. As requested, we performed additional functional experiments to support our proposed mechanism, specifically we have verified the functional importance of the NorD linker region. We believe that these experiments not only support our suggested mechanism but also help to clarify it. We have also made extensive revisions to the order and text to further clarify our points. Below is our detailed response to each of the reviewers' points.

REVIEWER COMMENTS

Reviewer #1 (Remarks to the Author):

In this manuscript, Kahle et al. determine the structure of the MoxR AAA+ ATPase NorQ and its partner VWA protein NorD using cryoEM. NorQD have been shown to facilitate iron insertion into cNOR (cytochrome C -dependent nitric oxide reductase); cNOR is involved in nitrate respiration. The authors show that a finger protruding from the VWA domain of NorD inserts into the central pore of the NorQ hexamer. The authors also show that the the C-terminus of NorD interacts with a post sensor 1 loop in NorQ. The authors obtain the cryoEM structure of NorQD from *Paracoccus denitrificans* and from *Jhaorihella thermophila* with the latter being at higher resolution. The structures are interesting. The structural data are further verified using mutational analysis, interaction studies, and ATPase assays. The authors propose a mechanism whereby NorQD 'twists' and stretches the NorD linker to enable metal insertion into its target cNOR.

While the structures are interesting, there is no evidence for the proposed action of NorQD on cNOR. In other words, the authors do not carry out experiments to demonstrate the proposed "twisting" mechanism. In the absence of such data, the manuscript is basically showing a series of structures of NorQD.

Response: As specified by the reviewer above, our structural data are supported by functional data, so we do not agree with the assessment that we are 'basically showing a series of structures'. Nevertheless, we do agree that the final model for the NorQD mechanism of action was partly speculative, and we have carried out a series of new experiments aimed at verifying the importance of the presumably unstructured linker region between the two folded domains of NorD. This data, now presented in a new Figure 3e, support the importance of the length and specific regions of the linker for NorQD-mediated Fe_B insertion into cNOR. We have also included a new Supporting Figure 13 of the AlphaFold-modelled interaction between NorQD and cNOR, supporting the general interaction mode via NorD. We have also clarified which parts of the final mechanism in Fig 5 remain partly speculative when concerning binding to the target protein cNOR.

1. The abstract is poorly written with no explanation of the function of NorQD. The abstract concludes with a proposed function of NorQD on cNOR, which is not experimentally addressed in the manuscript.

Response: We have rewritten the Abstract to include the function of NorQD and specify our findings as well as our proposed model.

2. Since the JtNorQD is obtained at higher resolution than that of PdNorQD, it might be better to start the manuscript with describing the JtNorQD structure.

Response: We agree with the reviewer that it makes sense to start with the structures of highest resolution and that is exactly what we did, as we have now clarified in the manuscript. The *Pd* NorQD structure is resolved to 3.2Å, while the *Jt*NorQD is resolved to 3.3Å. We wanted to build the manuscript up from building blocks of increasing complexity in order to aid understanding, starting from NorQ alone, adding the VWA domain and finally adding also the N-terminal domain.

3. In line 74, the authors should provide an explanation as to why the Walker B mutant was used for cryo-EM throughout the paper. They did say the mutant has much lower ATP hydrolysis, but they did not say why they did not use wild type for cryoEM. They could have used ADP.

Response: We have clarified in the manuscript why the Walker B mutation was used for all structures. It leads to better expression, and more stable complex formation with the substrate of the AAA+ protein (in our case the partner protein NorD). This is also seen in other AAA+ proteins [1, 2].

4. Line 81/82: ‘Bottom’ and ‘top’ are relative terms. When describing the subunits, it might be better to use the labels presented in the figures, i.e. subunits A-F. Similarly, throughout, when describing the seam subunits, refer to them by their label as well, to increase clarity.

Response: We have revised the manuscript to include the subunit labels for better clarity.

5. Line 109: Why was excluding ATP useful in obtaining this structure? Why would it have affected the resolution?

Response: When adding ATP to the *Pd*NorQNorD-VWA sample just before freezing grids, the complex falls apart to varying degrees, presumably affecting the resolution. Excluding ATP (purified ‘as is’) gave the best quality data.

6. Line 109: The structure only includes the VWA domain of PdNorD; the other domains/regions might play a role in the interaction as well, so it would be good to introduce the complex with the understanding of this limitation.

Response: We have re-written this section to clarify the composition of the complex and to highlight the missing N-terminal segments of NorD.

7. On line 118, the ‘finger’ was suggested to aid in nucleotide exchange, but activity assays showed greater ATPase activity without this finger. These data conflict with the earlier statement. How may they be reconciled?

Response: We have deleted the statement that the finger might aid in nucleotide exchange. This speculation did not prove to be correct as the reviewer correctly pointed out.

8. Does the disulfide bridge play any biological role, and has it been observed in other homologs?

Response: As stated on lines 252-260 in the discussion, such a disulfide has not been observed in other AAA+ ATPases. The biological role is hitherto unknown, but we suggest it could serve a redox regulation role. We hope to investigate its role in future studies.

9. Supplementary table 4 indicates assays were conducted in DTT, but that was not sufficient for full activity as in supplementary table 6. Therefore, all the other constructs and mutants also need to be purified with DTT.

Response: We agree and for precisely this reason Supplementary Table 4 states that all proteins were purified in the presence of 10mM DTT. We have also added that there is 2mM DTT in the measurement buffer.

10. In Supplemental Figure 4, the subsets 2 and 3 of have low cFAR values, which can indicate some extent of orientation bias. Perhaps the authors can comment on how confident they are regarding the biological relevance/the conclusions made on the two subsets. This puts into questions some of the mechanisms they are proposing.

Response: We have now highlighted in the manuscript that the PdNorQD complexes generally display preferential orientation in the ice, which we partly have overcome by collecting tilted data. As the reviewer correctly points out, subsets 2 and 3 of the ATP containing, reduced *PdNorQ^{WB}NorD^{VWA}* dataset show orientational bias. However, as can already be seen at much lower resolution than 6.6Å, the AAA+ rings are displaying high flexibility at the spiral seam so that NorQ subunits might be lost and NorD binding is impaired. No further, more detailed conclusions have been drawn from this dataset. We are not aware that above mentioned subsets 2 and 3 have any connection to our proposed mechanism.

11. Line 163, when reporting RMSD values, authors should mention if it is using all atoms of the relevant amino acids, or all heavy atoms, or just alpha carbons etc.

Response: Thank you for pointing this out. We have now clarified that we are generally using CA Atoms for RMSD determination.

12. Explain the dotted circle in the panels of Figure 1b.

Response: We thank the reviewer for spotting this mistake, we have now explained that the dotted circle outlines the MIDAS motif of the VWA domain.

13. The discussion could more explicitly acknowledge the limitations of the current data, especially the lower resolution in the VWA domain region and the lack of direct visualization of remodeling, and should consider alternative models.

Response: We have performed additional experiments to support our model and we have also clarified in the revised text which parts of our suggested model are supported by experiments.

14. Figure legends could be more detailed, especially regarding the number of replicates and statistical tests for the biochemical assays presented.

Response: We have added these details.

15. Although there is an overall schematic (Figure 5), it could benefit from more detail or a zoomed-in feature highlighting the proposed mechanism.

Response: We have revised the figure, see also response to reviewer 3.

Reviewer #2 (Remarks to the Author):

This manuscript reports cryo-EM structures of the MoxR AAA+ ATPase NorQ in complex with its VWA domain partner NorD. NorQ and NorD are essential for insertion of an Fe into the membrane bound cytochrome c-dependent NOR (cNOR). Coupled with ATPase measurements, the authors propose a structural model for how the NorQD complex induces structural changes in cNOR. The structures are compelling and shed light on an essential biological function. The manuscript should be published.

Response: We thank reviewer #2 for the very positive assessment of our manuscript.

Reviewer #3 (Remarks to the Author):

Kahle and colleagues present multiple structures of the AAA+ ATPase, NorQ, alone or with its VWA domain partner, NorD. NorQB structures were determined of both *Paracoccus denitrificans* and *Jhaorihella thermophila* homologs, and in oxidizing or reducing conditions. The JtNorQD cryo-EM data analysis resulted in four unique states, including one with only the VWA domain of NorD shown, and three states with unique NTD conformations of NorD. The authors conclude that these states represent the rotation of the VWA and NTD, leading to ATP dependent opening of cNOR and insertion of an iron ion.

The cryo-EM and image analysis was done to a high standard and well documented.

Response: We thank reviewer #3 for the general positive assessment of our data.

Major points:

1. The order in which these structures are presented seems backwards of a logical flow of information.

It is unclear why the authors do not include a reducing agent in their protein purifications, experiments, and cryo-EM samples in the first place. The way the manuscript is written, it seems like an oversight that is then corrected by redoing the experiment. If there is a

biological reasoning behind this, as is alluded to on Lines 258-260, please introduce it earlier. You could also start with the structures containing DTT, and then introduce the disulfide bridge found in the oxidized structure after. It seems that with an RMSD of ~1Å between them, your conclusions in the results section won't change much with this reorganization, and you can say that the under oxidizing conditions the ATPase activity is reduced, likely due to this disulfide forming.

Response: The reason we presented the oxidised structure first is indeed somewhat historical, since we have used the same buffer conditions in previous works. Only when we identified the disulfide bridge, we realized that the buffer conditions were influencing the ATPase activity. The structure of PdNorQ^{WBD}VWA under oxidizing conditions has the highest resolution, and the detailed interaction analysis was made using this structure. However, we agree with the reviewer that the logical flow improves if we introduce the reduced structure earlier, and we have reordered the manuscript this way. We now also refer to the discussion on the possible role of the disulfide bond in the Results section.

I feel similarly about the current order of starting with describing the Pd NorQD structures then transitioning to Jt NorQD. It seems that state 1 of the JtNorQD structures is the same as PdNorQD (RMSD 0.687Å). The interesting results are coming from the conformational switching between the NorD VWA/NTDs. It seems that the reason for this ordering is because there is significant anisotropy in the JtNorQD structures, limiting ability to solve the structures from those maps alone. I think there is nothing wrong with this, but the authors could do a better job creating a logical flow from one set of structures to the next.

Response: As stated above in response to reviewer #1, we have presented the data in the order of 'simple' to more complex to aid the understanding. The resolution of the PdNorQ^{WBD}VWA structure is higher than that of the JtNorQD complex and all our mutational studies are done on the Pd complexes. Hence, many of the functional data rely on the Pd complexes, while the Model of NorQD action is derived mainly from the Jt complexes, as these structures were unattainable using Pd complexes. We have now restructured the manuscript and tried to create a logical flow introducing each of the experiments.

2. The significance of the spiral staircase change in NorQD structures is unclear.

Starting with the title on line 90, the authors describe the staircase of NorQ as having a "steeper spiral staircase" when bound to NorD, as compared to the structure without NorD. In Line 101 authors say the ATP "complexes display a significantly steeper spiral staircase and stronger seam than seen in the 'as isolated' PdNorQWB complex". What do you mean by 'as isolated'? Is this the previous structure? Be clear here, as you use this wording a few more times.

Response: Thank you for pointing out this lack of clarity. We now state that the as isolated complex refers to the oxidized variant of complexes and refer to any additives during experiments by naming them directly (eg DTT, ATP). In Line 101 we were comparing PdNorQ^{WBD}VWA complexes obtained in the presence of ATP with PdNorQ^{WB} complexes without any added ATP. Both complexes were obtained in the presence of DTT.

Define 'significance' in the change in the staircase? Can it be described in a distance change? Unfortunately, the figures do not provide enough information about this change. Rather than showing the maps with slightly altered protomer E in Supplementary Figure 2, you should have the atomic structures of subunits of F, E and D overlaid so we can see their movement

down relative to each other. Perhaps it would be more appropriate to show different subunits- it depends which subunit to which you are aligning the structures.

Response: We have now added guidelines and distance marks to Supplementary figures S2a, S2c and S6c to highlight the change in the spiral staircase arrangements. In our hands, the change is best seen, when comparing the maps. An overlay of the atomic structures, even when only showing Ca traces, is very overwhelming. In AAA+ proteins, the helical shift between subunits is mostly dependent on the nucleotide bound to the subunits. Hence, the shift in the ATP bound subunits A-C is similar, but differs for the ADP bound subunits D-F as the reviewer points out correctly. It therefore makes most sense to align all spirals on the topmost subunit in the spiral, which is subunit A. And this is what we have done. We hope that the changes we introduced could clarify this point.

In Lines 102-105 the authors also describe the loss of subunits in the ring. The maps in Supplementary Figure 2 look like there is missing density in part of the subunits E and F, which likely implies flexibility and heterogeneity of particles in your data processing. Knowing this, it is hard to compare the “movement” of these subunits, as some may appear lower in the staircase compared to their counterparts that are entirely lacking that part of the subunit.

Response: The reviewer correctly points out that there is flexibility in the bottom most subunits of the *PdNorQ*^{WBD}*VWA* subsets (subunit E in subset 2; subunit F in subsets 1 and 3). But subset 2 also lacks subunit F completely, leading us to suggest that subunits can be lost, when all subunits in the complex are bound to ATP. Since the small and large ATPase subdomains of AAA+ complexes move as rigid bodies and the densities for the large subunits are resolved well enough to place a PDB model, the spiral staircase of complexes can be compared well.

3. The order of progression between states in *JtNorQD* is confusing and hard to understand from the figures and movies.

When you are comparing an ensemble of models of a homohexameric complex it can be unclear the order in which they progress in the mechanism, or how to assign the ‘top’ subunit for alignment. When watching the movies, it seems that the VWA domain is rotating a ton, with little movement of the ATPase. I wonder what the progression would look like if you instead aligned the structures based on the VWA domain to see how the subunits of *NorQ* moved relative to it. *NorQ* is the ATPase and is known to go through large conformational switching while processing substrates. It seems more likely it would be moving, and the rotation of VWA would be in response to this. Either way, the authors need to find a better way to demonstrate these conformational changes in their figures and movies. The current angles shown in Figure 3 are confusing, and you should find a better angle than a 90-degree rotation so we can understand that swing of the VWA domain.

Response: In our initial movie, we aligned the structures to the finger motif of the VWA domain and showed how all other domains (including the AAA+ subunits) moved relative to it. Now, we have revised the movie comparing all *JtNorQD* states such that we kept the subunit bound to the *NorD* N-terminal domain static so that all other domains move relative to it. We have also labeled the linker attachment points in *NorD* for more clarity. As can be seen very well in the movie, there is no movement between state 3 and state 4 apart from the

flipping back of the NorD VWA domain. All other structural features are identical. Dynamic movements, however, can be depicted in various ways relative to each other. One could argue that in the NorQD system, the NorD N-terminus is bound to a target on the membrane as well as the NorQ ring and the rest of the system moves relative to this attachment point. Hence, we kept the N-terminal domain static showing this scenario of events. To us it is clear how to assign the top subunit for alignment. The top subunit will always be the first ATP bound AAA+ domain in the spiral, next to the linker domain of NorD.

4. Proposed mechanism includes the substrate, cNOR, but the manuscript does not provide direct evidence for how NorQD interacts with this substrate.

The structures presented in this manuscript lack the membrane-bound substrate, cNOR. In Figure 5 they include cNOR and provide a hypothesis of how these structures would interact with that substrate. Is there previous data to suggest that NorQ does not translocate a part of cNOR into the central channel? If so, would both NorD and cNOR be accommodated in the channel? What is the evidence that NorQD would sit atop of cNOR in this way? The authors need to provide better justification for this hypothesis, which currently has no evidence from the data presented.

Response: It is true that there is no direct data on the interactions between NorQD and cNOR. As discussed in our previous publication [3] the chaperones presumably act on an assembly intermediate of cNOR and direct interactions are hard to study as we do not know the state of cNOR involved. We do, however, have indirect data showing that FeB insertion into cNOR depends on the activity and structural integrity of NorQD for several constructs (including the new NorD linker deletions described in the response to reviewer #1 above). We have now rewritten the discussion to clarify the evidence for our model. In detail, they are:

- a) The NorD finger occupies the central pore of the NorQ hexamer where other AAA+ proteins bind to target protein, making direct interaction with the cNOR target here unlikely. We have also added an AlphaFold model of the entire NorQ hexamer/NorD/cNOR complex to the Supporting information. We do not intend this model to give details about interactions as it has regions of very low confidence, but it shows a general interaction mode we propose in our model whereby the NorQD complex interacts with cNOR via the NorD only.
- b) We do know from our previous publication [3] that the MIDAS residues of the VWA domain of NorD are not needed for interaction with NorQ, but that they are essential for target remodeling. This indicates that they are involved in target (cNOR) binding, as shown for many other VWA domains.
- c) We have added new experimental data on the NorD linker region, which we present in the new Figure 3e. They show that deleting specific regions of the linker, which we selected based on their presumed roles as deduced from the alphaFold model (also shown in new Supplemental Figure 13), yields a NorQD complex unable to perform FeB insertion into cNOR. This supports our suggestion that the conformational changes in NorQD are leading to target remodelling that involves stretching the NorD linker. This data supports the general importance of the linker.

Minor points:

Some of the figures could use some adjustments for clarity.

1. In Figure 2, please show both residues involved in the bonds, including in the backbone. The way the structure is being displayed makes it hard to see the atoms involved in the interaction.

Response: Thanks for the suggestion. For more clarity in the figure and more specificity in presenting the data, we have opted to include the interactions between $PdNorQ^{WB}$ and $PdNorD^{VWA}$ in form of a table in Supplementary Table 3 & 4, including the interacting atoms and bond length.

2. In the Figure 3 legend you need to include a description of what colors associate with which domain. It is not clear here which domain is VWA or NTD of NorD.

Response: We are using the same color codes throughout the manuscript, depicted in Figure 1. We do agree with the reviewer however that it is beneficial to repeat/describe them in each caption, which we have done in the revised manuscript.

3. In Figure 5 you should add arrows between states. Would be helpful to have the structure/state on the figure itself, rather than just listed in the figure legend.

Response: We have revised the figure as suggested by the reviewer.

The methods section was written rather poorly, including what appear to be typos, informal writing style and incorrect shorthand. Please edit the methods section to have a similar writing style as the rest of the manuscript and correct the mistakes.

1. Line 638 – “BL21 DE. Cells” – do you mean BL21 DE3?

Response: Yes, fixed.

2. Line 640 – When describing protease inhibitor tablets it was called a “pill” on Line 640 and 652 but called a “tablet” to describe it on line 681. Be consistent.

Response: Fixed.

3. Line 643- Authors claim to use a “HiTrap” column for purification but use an imidazole gradient. Do you mean a “HisTrap”?

Response: Yes, fixed.

4. Line 737 – To describe the total number of particles, the shorthand “3 mio.” was used. Do you mean million? Either write it out or find a more appropriate shorthand.

Response: Yes, fixed

5. An example of informal writing can be found on Line 772 – “two trash classes”. What do you mean by trash classes? Find a better way to describe your process for eliminating bad classes.

Response: Yes, fixed.

1. Puchades, C., C.R. Sandate, and G.C. Lander, *The molecular principles governing the activity and functional diversity of AAA+ proteins*. Nature Reviews Molecular Cell Biology, 2020. **21**(1): p. 43-58.
2. Wendler, P., et al., *Structure and function of the AAA+ nucleotide binding pocket*. Biochimica et Biophysica Acta (BBA)-Molecular Cell Research, 2012. **1823**(1): p. 2-14.
3. Kahle, M., et al., *Insights into the structure-function relationship of the NorQ/NorD chaperones from Paracoccus denitrificans reveal shared principles of interacting MoxR AAA+/VWA domain proteins*. BMC biology, 2023. **21**(1): p. 47.

Response To Reviewers; NCOMMS-25-24940A

Reviewer #1 (Remarks to the Author):

The authors made some effort to make the suggested changes. They do additional experiments with truncations of the linker domain that essentially show inactive protein, while still being able to form the complex. This is to try to address the lack of experimental data to support their model. They explained away other issues I raised and they made all the clarity changes I wanted for the text, but not really to the model figure. For that figure, they just added directional arrows and a couple of labels.

From previous review round:

15. Although there is an overall schematic (Figure 5), it could benefit from more detail or a zoomed-in feature highlighting the proposed mechanism.

Response: The reviewer raises concerns about the schematic Figure 5, in which we propose how our structural and biochemical data can be integrated into a mechanism of cNor remodelling by NorQD. In the absence of structural data of the cNor/NorQD holo-complex, we hypothesise how the structures described in this manuscript fit to our biochemical data and previously published findings. With this in mind, we suggest that the observed twisting motion of the NorD VWA domain with regard to its N-terminal domain translates into remodelling action on cNor and iron insertion into NorB. We revised Figure 5 again to highlight the hand-over-hand mechanism in NorQ that twists the NorD VWA domain as well as the positioning of linker attachment points and MIDAS motif that indicate remodelling of the linker domain. We have also added a zoomed-in feature, as suggested by the reviewer, that shows a view onto the NorQD complex as seen from the membrane to illustrate the distance change in the linker attachment points. We furthermore indicate which states of NorQ exchange nucleotide.

Reviewer #2 (Remarks to the Author):

The authors have adequately addressed the reviewer comments.

Reviewer #3 (Remarks to the Author):

The significance of the spiral staircase change in NorQD structures is unclear

2. Starting with the title on line 90, the authors describe the staircase of NorQ as having a “steeper spiral staircase” when bound to NorD, as compared to the structure without NorD. In Line 101 authors say the ATP “complexes display a significantly steeper spiral staircase and stronger seam than seen in the ‘as isolated’ PdNorQWB complex”. What do you mean by ‘as isolated’? Is this the previous structure? Be clear here, as you use this wording a few more times.

Define ‘significance’ in the change in the staircase? Can it be described in a distance change? Unfortunately, the figures do not provide enough information about this change. Rather than showing the maps with slightly altered protomer E in Supplementary Figure 2, you should have the atomic structures of subunits of F, E and D overlaid so we can see their movement down relative to each other. Perhaps it would be more appropriate to show different subunits- it depends which subunit to which you are aligning the structures.

Author Response: We have now added guidelines and distance marks to Supplementary figures S2a,S2c and S6c to highlight the change in the spiral staircase arrangements. In our hands, the change is best seen, when comparing the maps. An overlay of the atomic structures, even when only showing Ca traces, is very overwhelming. In AAA+ proteins, the helical shift between subunits is mostly dependent on the nucleotide bound to the subunits. Hence, the shift in the ATP bound subunits A-C is similar, but differs for the ADP bound subunits D-F as the reviewer points out correctly. It therefore makes most sense to align all spirals on the topmost subunit in the spiral, which is subunit A. And this is what we have done. We hope that the changes we introduced could clarify this point.

Reviewer Response: Authors still have not defined what they deem as a ‘significant’ change in the staircase. Please include comments in text about the distance change.

I disagree that having an overlay of the atomic structures would be overwhelming. You do it in Supplementary figure S2e. You could make a similar figure by showing the atomic structure of subunit E as an overlay of the “as isolated” with each of the ATP structures, as they are currently aligned in figure S2 a and b.

Response: We have now included the increase in the total helical rise between subunits A to D for PdNorQ^{WB} - PdNorQ^{WB}D^{VWA} (line 126) and subunits A to E for PdNorQ^{WB} - PdNorQ^{WB}D^{VWA} (ATP, subset 1) (line 143) in the

manuscript. Based on the reviewers request to “make a similar figure by showing the atomic structure of subunit E as an overlay of the “as isolated” with each of the ATP structures, as they are currently aligned in figure S2 a and b”, we have added a new Supplementary Figure 6c showing an overlay of all relevant *Pd* complexes to visualise the changes in helical rise. We would like to point out that the increase in total helical rise is mainly due to shifts in subunits C to F, as ATP binding only affects subunits D to F (in complexes in the presence of ATP). Furthermore, Figures S2a and b are not showing aligned structures, but only *PdNorQ^{WB}* as isolated. All models shown in Supplementary Figure 2a-d are already aligned to subunit A of NorQ to allow for visual comparison of the helical staircase in the complexes. We have now also stated that in the legend of this figure.

3. The order of progression between states in JtNorQD is confusing and hard to understand from the figures and movies.

When you are comparing an ensemble of models of a homohexameric complex it can be unclear the order in which they progress in the mechanism, or how to assign the ‘top’ subunit for alignment. When watching the movies, it seems that the VWA domain is rotating a ton, with little movement of the ATPase. I wonder what the progression would look like if you instead aligned the structures based on the VWA domain to see how the subunits of NorQ moved relative to it. NorQ is the ATPase and is known to go through large conformational switching while processing substrates. It seems more likely it would be moving, and the rotation of VWA would be in response to this.

Either way, the authors need to find a better way to demonstrate these conformational changes in their figures and movies. The current angles shown in Figure 3 are confusing, and you should find a better angle than a 90-degree rotation so we can understand that swing of the VWA domain.

Author’s Response: In our initial movie, we aligned the structures to the finger motif of the VWA domain and showed how all other domains (including the AAA+ subunits) moved relative to it. Now, we have revised the movie comparing all JtNorQD states such that we kept the subunit bound to the NorD N-terminal domain static so that all other domains move relative to it. We have also labeled the linker attachment points in NorD for more clarity. As can be seen very well in the movie, there is no movement between state 3 and state 4 apart from the flipping back of the NorD VWA domain. All other structural features are identical. Dynamic movements, however, can be depicted in various ways relative to each other. One could argue that in the NorQD system, the NorD N-terminus is bound to a target on the membrane as well as the NorQ ring and the rest of the system moves relative to this attachment point. Hence, we kept the N-terminal domain static showing this scenario of events. To us it is clear how to assign the top subunit for alignment. The top subunit will always be the first ATP bound AAA+ domain in the spiral, next to the linker domain of NorD.

Reviewer response: Authors say that it is clear to assign the top subunit as the first ATP bound AAA+ domain in the spiral. I agree, and I can see that is how they have assigned it in Supplementary Figure 11. But, since the movie is aligned to the NTD, this is changing how the other subunits are assigned.

In the Supplementary figure 11, the NTD is bound to subunit B in state 2, and subunit C in state 3. Now that you have aligned them to the NTD in the movie, in the transition between state 2 and state 3 we see the NTD bound to subunit B in both, and therefore now we see subunit F go from being a seam subunit to the new top of the spiral staircase in this transition, with subunits D, E and F all moving in an upward direction as a rigid body. Have you tried aligning to the NTD and morphing from state 3 to state 2? It seems like you might have it backwards, based on what we know of AAA+ motor translocation mechanisms.

Evidence that the motor moves in a hand-over-hand mechanism can be found in numerous studies. It is particularly easy to see in AAA+ motor structures of the 26S proteasome, which has a heterohexameric AAA+ that can be aligned to the 20S core peptidase; therefore, movement of each subunit can be followed through the ATP hydrolysis cycle. In Dong et al. Nature 2019 and de la Peña et al. Science 2018, they saw consecutive ATP hydrolysis and translocation states in which the ATP-bound subunits move downwards as a rigid body and the bottom subunit releases substrate and rebinds at the top of the spiral. Assignments of ‘top’ and seam subunits is done the same way as the authors described in this study.

I agree with authors that you need to pick something to align to, and in your study it makes sense to align all the

models to the NTD, since that is likely bound to a membrane. This would require more self-consistency in the manuscript. You should align your models to the NTD in Supplementary Figure 11a and b. You should also try morphing from state 3 to state 2 and see if that direction of movement is more consistent with previous studies, in which rigid body movement of the ATP-bound subunits is downward, and the seam unit will move up to become the new 'top' subunit. This could show an interesting result of how the movement of the NTD and VWA is related to a translocation event.

Response: We are happy to read that the reviewer agrees with how we have aligned the complexes. In fact, the alignments in Figures 3c, S2a-d, S4d, S6, and S11a-b are done by aligning on subunit A of NorQ (always the top ATP bound subunit in the spiral) while Figure 4 and the movie shows the complexes being aligned on the NTD of NorD, highlighting the hand-over-hand mechanism of NorQ. We deliberately chose the different perspectives of alignment to point out the similarities between NorQ states (alignment on subunit A) or the action of NorQD relative to a fixed membrane interaction (alignment on NTD). To facilitate understanding of the movements and the different perspectives, we consistently colored the NorQ ring with shades of blue from the topmost subunit in the spiral (A, dark blue) to the subunit at the seam (F, light blue).

Regarding the movie and Figure 4, we propose the sequence of events, because NorQ exactly follows a hand-over-hand mechanism as seen in other AAA+ clades. For clarification, we now explicitly state that NorQ is working according to the hand-over-hand mechanism (line 295, legend Fig 5). As our work demonstrates, clade 7 AAA+ proteins do not seem to use the hand-over-hand mechanism on substrate being moved through the central pore, but to rotate the VWA protein partner in the pore. We do not think the reviewer's suggestion to morph from state 3 to state 2 would increase clarity. As shown in the figure below, a transition from state 3 to state 2 would imply that the top ATP bound subunit (subunit A) in the spiral would have to lose ATP and the apo subunit (subunit F) would have to bind ADP - something that has never been observed before in any other AAA+ complex.

To clarify the model and avoid any misunderstandings, we have revised the movie to show the change in color code upon ATP binding and reaching up of the seam subunit. We also explain the individual snapshots in more detail and clearly separate the view sequences now.